# Light-oriented 3D printing of liquid crystal/photocurable resins and in-situ enhancement of mechanical performance

Xiaolu Sun[1,2,3,4,5], Shaoyun Chen [1,2,3,4,5] ✉, Bo Qu[1,3,4], Rui Wang[1,3,4], Yanyu Zheng[1,3,4], Xiaoying Liu[1,3,4], Wenjie Li[1,3,4], Jianhong Gao[1,3,4], Qinhui Chen [2] ✉ & Dongxian Zhuo [1,2,3,4] ✉

Additive manufacturing technology has significantly impacted contemporary industries due to its ability to generate intricate computer-designed geometries. However, 3D-printed polymer parts often possess limited application potential, primarily because of their weak mechanical attributes. To overcome this drawback, this study formulates liquid crystal/photocurable resins suitable for the stereolithography technique by integrating 4'-pentyl-4-cyanobiphenyl with a photosensitive acrylic resin. This study demonstrates that stereolithography facilitates the precise modulation of the existing liquid crystal morphology within the resin. Furthermore, the orientation of the liquid crystal governs the oriented polymerization of monomers or prepolymers bearing acrylate groups. The products of this 3D printing approach manifest anisotropic behavior. Remarkably, when utilizing liquid crystal/photocurable resins, the resulting 3D-printed objects are approximately twice as robust as those created using commercial resins in terms of their tensile, flexural, and impact properties. This pioneering approach holds promise for realizing autonomously designed structures that remain elusive with present additive manufacturing techniques.

Additive manufacturing (AM) technology differs from traditional equal- and reduced-material manufacturing methods as it quickly and easily produces materials of diverse designs[1–3]. Therefore, three-dimensional (3D) printing has become one of the main approaches for developing future product processing methods[4–7]. This promising technology has developed rapidly in the last few years and has found numerous applications in various fields, ranging from personalized consumer products to biomedical engineering and the automotive and aerospace industries[5–9]. Hence, 3D printing is a key component and a paradigmatic example of the next industrial revolution[10]. Many AM technologies currently exist, such as stereolithography (SLA),

selective laser sintering (SLS), and fused deposition modeling (FDM)[11,12]. Among these, SLA is one of the most widely applied techniques because of its high precision and the requirement of only a light source to induce photopolymerization; therefore, this method is low in energy consumption and environmental pollution[13–16]. In addition, SLA is the only light-curing 3D printing technology that can print large models. The laser beam, under the action of a deflecting mirror, can scan on a liquid surface, and the trajectory of the scan and the presence or absence of light are controlled by a computer. The liquid cures at the point of luminous contact, manufacturing the parts by deposition speed up to $10^5$ mm³/h[17]. For further industrial progression,

¹College of Chemical Engineering and Materials Science, Quanzhou Normal University, Quanzhou, Fujian 362000, P. R. China. ²College of Chemistry and Materials Science, Fujian Normal University, Fuzhou, Fujian 350007, P. R. China. ³Fujian University Engineering Research Center of Polymer Functional Coating based Graphene, Quanzhou, Fujian 362000, P. R. China. ⁴Fujian Key Laboratory of New Materials for Light Textile and Chemical Industry, Quanzhou, Fujian 362000, P. R. China. ⁵These authors contributed equally: Xiaolu Sun, Shaoyun Chen. ✉e-mail: chshaoy@qztc.edu.cn; chenqh@fjnu.edu.cn; dxzhuo@qztc.edu.cn

developing high-performance photosensitive resin materials for 3D printing is crucial[18–26]. Previous studies have proposed three strategies to address this need: the use of different monomers[18–21], the addition of inert fillers or additives[22–24], and the adoption of an epoxy acrylate hybrid system[25,26]. Among these approaches, the addition of inert fillers or additives, such as silica fillers[27], calcium sulfate whisker fillers[28], graphene oxide fillers[29,30], polysiloxane core–shell nanoparticles[31], carbon-based nanoparticles[32], polyimides[33], and liquid crystal (LC) resins[24], is the most convenient and common method because it reduces volume shrinkage and enhances the mechanical performance of the resulting object. Modifying the additive particles and grafting certain photosensitive groups can enhance the compatibility between the additive particles and the photosensitive resin, effectively improving the mechanical or thermal properties of final 3D-printed products. For example, Lin et al. [30] found that adding 0.2% graphene oxide to the SLA precursor increased the tensile strength and ductility of the printed composites by 62.2% and 12.8%, respectively. The enhanced ductility of the polymers was attributed to the presence of graphene oxide, which increased the crystallinity of the composite.

LCs are an intermediate phase between the solid and liquid states of matter and show strong nonlinear optical effects. LCs have the molecular orientation properties of a solid crystal but change shape as a fluid[34–37]. LCs are tightly and stably enclosed within polymers for use in applications, providing a compact, reliable, and easy-to-use system[38]. The LCs and polymer are blended into a common solution during the phase separation procedure. Subsequently, the mixture is sprayed, and the LC molecules detach from the polymer, remaining in the medium as distinct LC droplets[39–41]. Polymer-stabilized LC (PSLC) and polymer-dispersed LC (PDLC) systems with specific properties can be produced by manipulating the microstructure of the LC/polymer complexes[42–51]. PDLC films are obtained by dispersing LC droplets in a polymer matrix. Polymerization is induced by methods such as thermal curing and UV irradiation. When these methods are applied, the solubility of the LCs in the polymer matrix decreases, causing phase separation and the formation of microphase-separated structures[52–58]. Electronically controlled LC composites respond only to electric signals; however, thermally controlled composites respond to both electric power and temperature changes within their environment[59–61].

LCs can respond to the effect of temperature, light, and electricity; therefore, this study proposes that during the SLA process, in the absence of photon absorption, the laser beam provides a photoelectric field that changes the molecular orientation during printing. To the authors' knowledge, there are no previous reports on the polymer orientation of products during the SLA process. Figure 1 shows the proposed mechanism of LC orientation during the SLA process. First, a certain proportion of LCs is added to the photosensitive resin and dispersed by shear dispersion (Fig. 1a). Subsequently, curing of the photosensitive resin is performed, during which the solubility of LCs in the polymer matrix decreases, resulting in the agglomeration of LC particles; thus, phase separation occurs. At this point, the curing time of each layer is controlled by adjusting the thickness of the print layer (Fig. 1b). Shortening the curing time of each layer results in the fixation of LC particles to the print layers before aggregation can occur. Thus, finding the right print layer thickness can prevent the aggregation of LCs and the generation of phase separation. Finally, the acrylate group of the photosensitive resin (Fig. 1c) will emit heat during the reaction process. Simultaneously, the laser power will produce a photoelectric field that will cause the LCs to transition to a specific orientation. This change will drive the polymer chains into the same orientation, aligning them in one direction.

One of the most well-known LC series of calamitic molecules is 4-cyano-4′-alkylbiphenyl (nCB), where n represents the number of carbon atoms present within the alkyl chain. Excluding 5CB and 6CB, the nCB series displays an odd–even trend of CN...CN and CN...phenyl interactions in all crystal structures. The odd–even effect in the

packing modes for higher homologs ($n \le 7$) is known to be CN...CN or CN...phenyl interactions. 4′-Pentyl-4-cyanobiphenyl (5CB) is one of the most studied LC materials owing to its readily accessible nematic range near room temperature that enables its use as a suitable model compound for studying the physical behavior of simple nematics[62–64]. The 5CB LCs studied in our experiments are linear LCs at room temperature (20–35 °C). Without photon absorption, a laser power of $10^2$–$10^3$ W/cm$^2$ is required to generate the photoelectric field that changes the molecular orientation[65]. The energy required for the photoelectric field to change the molecular orientation originates from a small shift in the optical frequency[66]. Preparation of LCs that retain their high-performing characteristics when used in a simple 3D printing technique—here, mixing 5CB with a photosensitive resin—will open a range of possible applications in industry[67–70]. Compared with the traditional use of photosensitive LC materials in 3D printing, this approach is more competitive[24,69]. For example, self-assembly methods containing materials such as copolymers, colloids, and LCs have created microstructures with tunable optical properties, providing many unique material systems for developing smart optical devices using AM methods[71]. In addition, by combining LCs with advanced 3D printing processes, smart materials with sensing capabilities can be fabricated based on the nematic and anisotropic phase transitions of LCs[72–74].

This study proposes a 3D printing strategy that uses a photosensitive resin with LC-controllable morphology. The strategy includes two design steps: first, 4′-pentyl-4-cyanobiphenyl (5CB) is blended with an acrylic-acid-based photosensitive resin to produce 5CB/photosensitive resin (PR-5CB); the dispersion and rheological properties of PR-5CB resins are characterized. Second, the SLA process is used to print PR-5CB resins, during which the printing direction and printing resolution of the stack are regulated by controlling the printing parameters so that the LC orientation in the resin is fixed. Once the single layer of resin is cured and the entire network system is crosslinked, the rod-like structure of the LCs is fixed. This study uses a combination of 3D printing technology and precise modulation of the LC morphology to meet the desired product needs. Compared with the use of conventional reinforcing materials, using LCs with controlled orientation is highly suitable for commercial use, offering a wide range of applications for the development of AM. Finally, the thermal and mechanical properties of the samples printed with each formulation of PR-5CB were studied to further investigate the enhancement mechanism of the printed samples obtained by the SLA printing technology.

## Results

### Dispersibility and rheological properties of PR-5CB resins

Currently, most of the common photosensitive resins in the market are various acrylates, such as epoxy acrylates, urethane acrylates, and polyester acrylates. A general SLA photosensitive resin formulation was selected, and photosensitive resins containing different quantities of 5CB (PR-5CB resins) were prepared using solution blending (Supplementary Table 1). Good dispersion is vital for obtaining high-performing 3D-printed samples using PR-5CB resins. Therefore, it is essential to study the rheological properties of the composites as influenced by additive content, as well as the effect of adding plasticizers and compatibility agents, which can improve feedback processability. In addition, if the additive is not well bonded to the resin matrix, it will lead to a degradation of mechanical properties in the cured product. For the PR-5CB resins, a well-mixed resin was observed under a polarizing optical microscope (POM), and many rod-shaped LC molecules were distributed throughout the field of view (Supplementary Fig. 1a). This indicates that 5CB was well-dispersed within the resin substrate and demonstrates the feasibility of obtaining the desired properties of the PR-5CB resin. Based on the POM calculations, the rod LCs were between 13 and 20 μm in size.

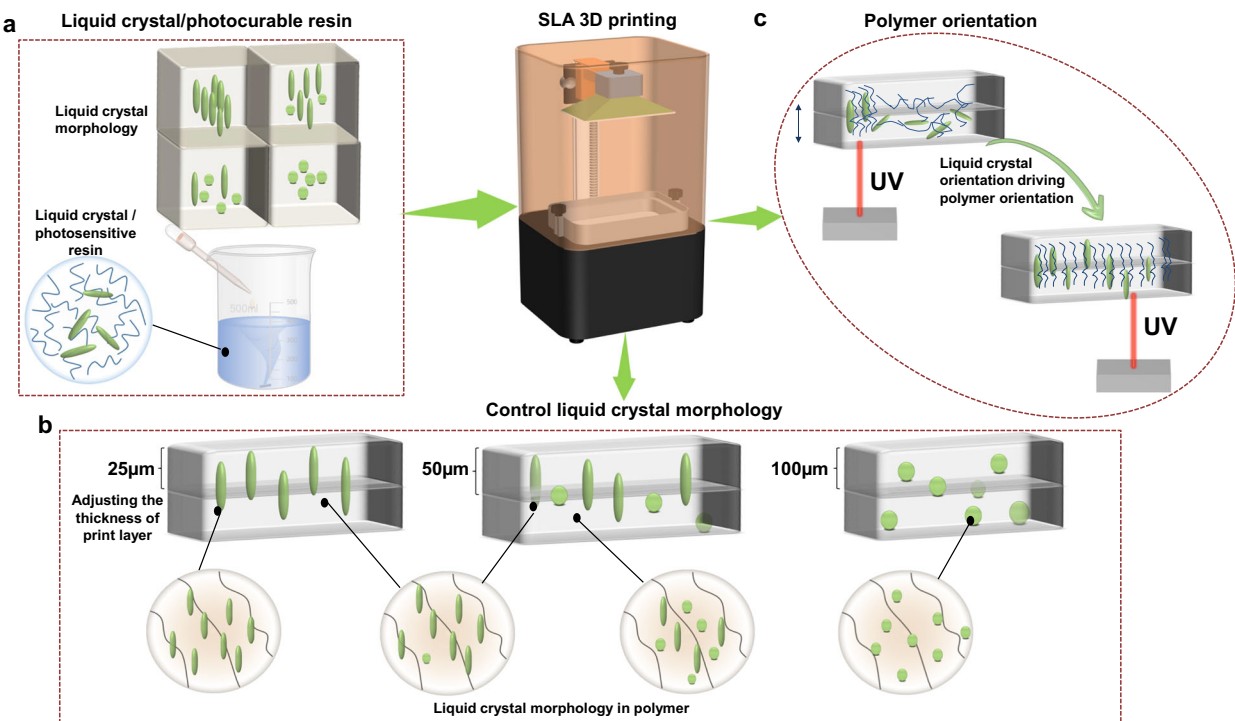

**Fig. 1 | LC orientation during the SLA 3D printing process. a** Preparation of LC/photosensitive resin. **b** Changing the print layer thickness to control the LC morphology in the polymer. **c** LC orientation driving the polymer orientation during printing.

In 3D printing, the resin must be sufficiently leveled or scraped before each layer of raw material is cured and set; this usually requires good flowability. Adding 5CB increases the system viscosity (Supplementary Fig. 1b), which is at its maximum (5.7 Pa·s) when the shear rate is 1 s$^{-1}$. As the shear rate increases, the overall viscosity of the system significantly decreases. This phenomenon is consistent with fluid shear thinning behavior. Under high-speed shear, the viscosity of PR-5CB (0.2 Pa·s) is 75% lower than that of commercial inks (0.8 Pa·s), providing the possibility for mass production using 3D printing. To further study the dynamic mechanical changes in the curing process of the PR-5CB resins (Supplementary Fig. 1c), UV light irradiation was implemented, and the PR-5CB resins were gradually cured. After 20 s, the energy storage modulus gradually increased; however, it remained constant once curing was completed. Therefore, the maximum energy storage modulus for the 3D-printed sample based on the PR-5CB resins was obtained. This marginally improved the light-curing speed of the PR-5CB-3 resin when compared with that of the commercial resin.

### Investigating the light orientation mechanism of PB-5CB resins by SLA 3D printing

Experiments were performed to verify the orientation of the 5CB LCs under UV laser irradiation. Samples were printed with the printing direction parallel to the tensile test direction (vertical printing) (PC-5CB//) and samples with the printing direction perpendicular to the tensile test direction (horizontal printing) (PC-5CB⊥), as shown in Fig. 2a. Figure 2b shows the POM images of the surface and cross-sectional area of the 3D-printed tensile test sample at the 0° and 45° directions. The surface and cross-section of the 3D-printed products without 5CB were always dark under the POM when the rotary stage was turned 360° (Supplementary Movie 1 and Supplementary Movie 2). This indicates that the sample was isotropic, regardless of whether it was printed along the vertical or horizontal direction. However, when the 5CB LCs were added to PR-5CB-3//, the surface observed using POM showed four light and four dark images (Supplementary Movie 3), whereas the cross-sectional POM remained dark (Supplementary

Movie 4), indicating that the PR-5CB-3// polymer was anisotropic. To further substantiate this result, PR-5CB-3⊥ was observed under the POM, and its surface was dark (Supplementary Movie 5); however, four of its cross-sectional POM images were light, while four were dark (Supplementary Movie 6). In a confined environment, the mobility of LCs is limited, compelling them to interact with their surrounding interface. This interaction dictates the orientation of the LC molecules in the absence of external influences. Typically, when the intermolecular force among LC molecules surpasses the interaction between the LCs and the interface, the LC molecules tend to orient themselves perpendicular to the surface. On the other hand, when the surface energy of the substrate exceeds that of the LC molecules, the molecules generally align parallel to the surface. The alignment pattern of the LC molecules can be ascertained using Fourier transform polarization infrared spectroscopy (FTIR). The -CN group present in the 5CB molecule, exhibiting a pronounced vibrational absorption around 2226 cm$^{-1}$, serves as an effective probe. Notably, the vibrational direction of the -CN aligns with the long-axis direction of the LCs. Consequently, by analyzing the polarized infrared absorption spectrum of -CN, one can determine the alignment orientation of LCs. Supplementary Fig. 2a displays the test films produced using different printing techniques, each having a thickness close to 150 µm. Post printing, these films were examined by FTIR. In this figure, P denotes the polarization direction of the IR polarized light, and E represents the vibration vector of the polarized UV light. A strong vibrational absorption of -CN was observed when P was parallel to E, and a diminished absorption when P was perpendicular to E (as illustrated in Supplementary Fig. 2b). This observation indicates that the LC molecules preferentially align in a direction congruent with the UV light vibration vector.

According to the original concept of this study−5CB is oriented during the printing process, and the other photosensitive resin fixes 5CB−bright lines should be observed only in a certain direction. However, the results indicate that the process of 5CB orientation also drives other acrylate prepolymers to orient along the orientation direction of 5CB. Therefore, the entire fixed 3D printing polymer is

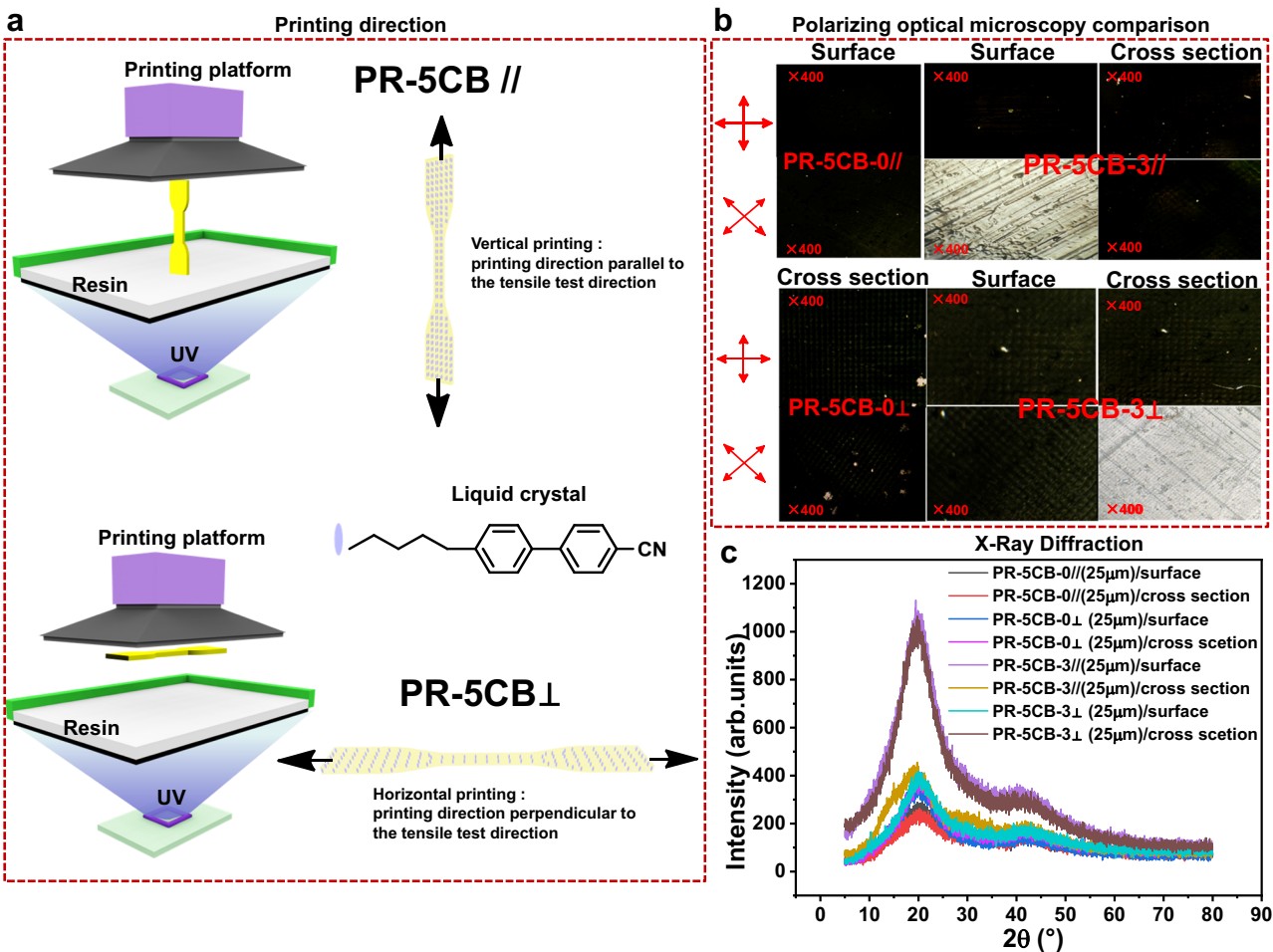

**Fig. 2 | Light orientation of PB-5CB resins by SLA 3D printing. a** Diagram showing the 3D printing orientation. **b** POM results for PR-5CB samples printed in vertical and horizontal directions at 0° and 45°. **c** XRD results for PR-5CB samples printed in vertical and horizontal directions.

anisotropic. It is also observed in the POM diagrams and schematics of the PR-5CB-3// surfaces and the PR-5CB-3⊥ cross-sections that the polymers are oriented along the direction of light, and 5CB is anchored between the 3D-printed layers. This is an important finding regarding LCs used for 3D printing. Typically, the LC prepolymers used to 3D-print LC elastomers have photosensitive groups (e.g., RM257). However, this study used small and unreactive 5CB LCs that can be oriented under UV laser irradiation to drive the orientation of the photosensitive resin. Thus, the results of this study are important for the development of light-cured 3D printing. To further verify the mechanism, the samples produced using different printing methods were subjected to surface and cross-sectional X-ray diffraction (XRD) tests (Fig. 2c). The results were in agreement with those of POM as follows: first, the low surface and cross-sectional XRD peaks of the PR-5CB-0// and PR-5CB-0⊥ samples were the same. This indicated that the polymer aggregation morphology of PR-5CB-0 was not greatly affected by the change in printing direction. In addition, the surface and cross-sectional polymer morphologies of the samples were the same, indicating that the polymer was not anisotropic. However, the surface XRD peaks of PR-5CB-3// samples were significantly higher than those of the cross-sectional XRD peaks, suggesting that the crystallinity of the surface and cross-sectional aggregates were different, as were their aggregation states. The cross-sectional XRD peak of the PR-5CB-3⊥ sample was significantly higher than its surface XRD peak. Both showed higher crystallinity along the UV irradiation direction than that of the other directions, implying that the orientation of the polymer was along the UV irradiation direction.

To corroborate the orientation of the LCs and to discern the influence of this orientation on the attributes of 3D-printed items, we evaluated the disparities in the mechanical properties of the two resins. The tensile and flexural properties of the 3D-printed products containing 3% 5CB LCs were higher than those of the samples containing no added LCs (Fig. 3a, b). In addition, the properties of the various stacking forms significantly differed after adding 5CB LCs; however, the mechanical properties of the resin samples without LCs were almost unaffected by their stacking forms. The tensile strengths for PC-5CB-0//, PC-5CB-0⊥, PC-5CB-3// and PC-5CB-3⊥ were 41.1, 43.5, 122.2, and 91.6 MPa, respectively (Fig. 3a), whereas the elongation at break values were 13.1%, 11.6%, 23.4%, and 14.6%, respectively. These results show that the mechanical properties of the conventional photosensitive resins of the isotropic PC-5CB-0 do not significantly change for products printed in different directions. However, the tensile strength and elongation at break for PC-5CB-3// were 1.3 and 1.6 times higher than those of PC-5CB-3⊥, respectively. Consequently, the mechanical properties of the products that were printed along the direction of layer stacking were better than those of the cross-sectional products. This suggests that LCs are ordered along the layer stacking direction, and the long-range ordered LCs improve the strength of the material. To further assess this conjecture, the samples were subjected to additional bending (Fig. 3b), impact strength, and hardness tests (Fig. 3c). The results showed that changing the 3D printing orientation of the PC-5CB-0 sample does not affect its mechanical properties. However, the flexural strength, flexural modulus, and impact strength of the PC-5CB-3// sample were 222.1 MPa, 2836.8 MPa, and 11.09 kJ/m²,

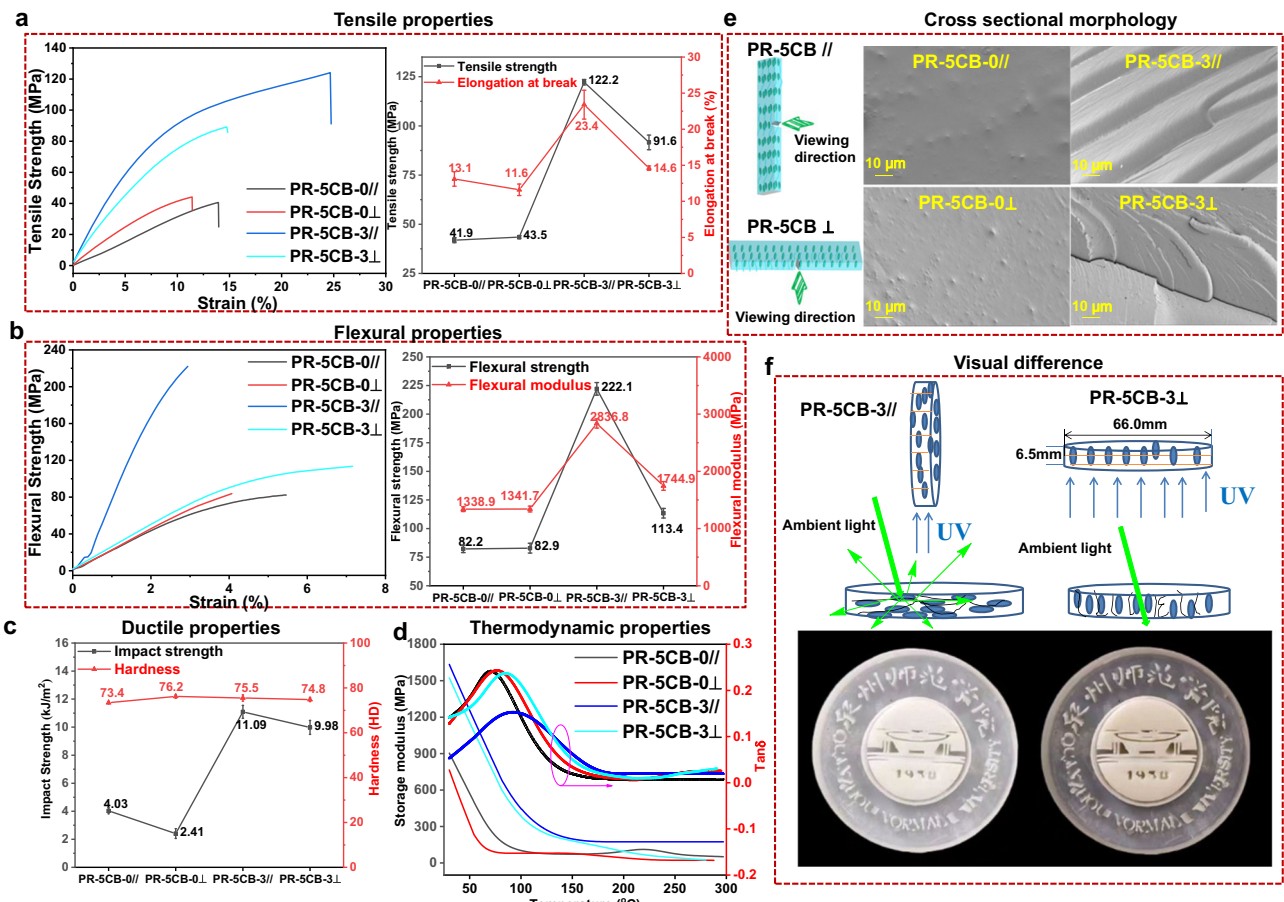

**Fig. 3 | Performance differences of 3D samples based on PR-5CB resins printed in vertical and horizontal directions. a** Stress−strain curves and tensile properties. **b** Bending strain curves and flexural properties. **c** Impact strength and hardness. **d** Storage modulus and loss factor curves. **e** SEM images of fracture surface morphology of the 3D-printed parts. The viewing direction is perpendicular to the printing direction for PC-5CB// samples, whereas the viewing direction is parallel to the printing direction for PC-5CB⊥ samples. **f** Visual difference in the printed model between the vertical and horizontal directions. All the data in (**a–c**) were collected five times, and the error bars represent the standard deviation.

respectively; these values were higher than those of the PC-5CB-0 sample, and 1.9, 1.6, and 1.1 fold higher than those of the PC-5CB-3⊥ sample. The storage modulus and loss factor of the PR-5CB resin-printed products with temperature was investigated using the DMA technique, and the results are shown in Fig. 3d. The storage modulus and $T_g$ values were 875.4 MPa and 71.7 °C, 765.2 MPa and 76.2 °C, 1702.3 MPa and 91.6 °C, and 1524.1 MPa and 85.5 °C for PC-5CB-0//, PC-5CB-0⊥, PC-5CB-3//, and PC-5CB-0⊥, respectively. Therefore, the expected trade-off between strength and toughness was not observed; the addition of 5CB improved the strength and toughness of the 3D-printed products. This was mainly owing to the similarity between the orientation of 5CB and the direction of light irradiation during sample curing.

The impact fractures of the printed products obtained from samples PR-5CB-0 and PR-5CB-3 were observed using scanning electron microscopy (SEM), and the results are shown in Fig. 3e. The PR-5CB-0 system had a smooth cross-section and a single direction for the microcrack owing to the poor molecular movement of the cured resin. This makes it difficult to produce yield deformation; additionally, the microcrack is essentially unhindered during expansion, which is typical of brittle damage. With the addition of 3% 5CB, roughness and river-like cross-sections appeared in the cross-section of PR-5CB-3. Furthermore, a branching extension of microcracks, which showed whitening, was evident, and the extension of microcracks ended with a typical river-like cross-section. Regarding the PR-5CB-3⊥ cross-section,

the surface granularity was clearer, and its appearance was also river-like. This is mainly because 5CB produces a one-dimensional orientation in the printing process, and the PR-5CB-3// specimen is cracked by impact. This occurs because the impact direction is perpendicular to the direction of 5CB and polymer orientation, resulting in the formation of a bridging zone in the fractured area, which prevents crack expansion and improves the impact strength of the material. Conversely, when the PR-5CB-3⊥ specimen was impacted, the 5CB and polymer orientation direction was parallel to the fractured surface. Therefore, the bridging zone could not be formed, resulting in a reduced impact strength enhancement effect compared to that of the PR-5CB-3// specimen. This result shows that, without the addition of 5CB, the difference in print orientation has little effect on the morphology of the cross-sectional fracture. However, the addition of 5CB effectively enhances the tough fracture of the 3D-printed product, and the change in print orientation affects the surface morphology of the fracture. The above analysis was consistent with the results of the mechanical property tests.

The LCs were oriented along the printing direction when the printing directions were vertical (PR-5CB-3//) and horizontal (PR-5CB-3⊥) (Fig. 3f). However, planar (flat) observations showed that the orientation of the 5CB LCs and polymer chains in the PR-5CB-3// samples were perpendicular to visible light; additionally, upon visible light irradiation, the 5CB and polymer chains would reflect the light, limiting penetration of the printed model by visible light,

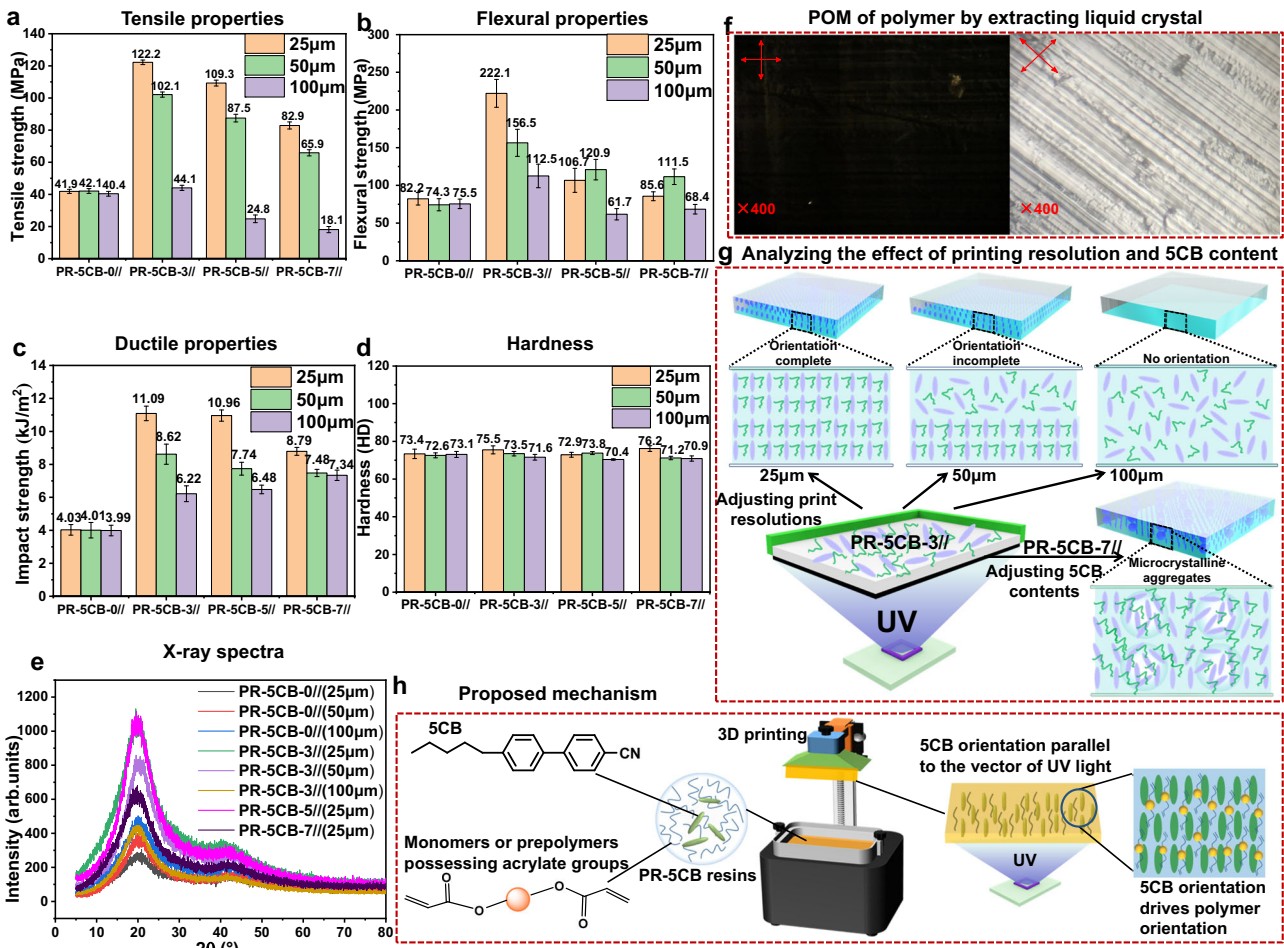

**Fig. 4 | Investigating the influence of the printing resolution and LC content and proposed orientation mechanism for photosensitive resins. a** Tensile strength. **b** Flexural strength. **c** Impact strength, **d** Hardness, **e** XRD of PR-5CB resins at different printing resolutions and contents. **f** POM of PR-5CB-3// (25 μm) after extracting 5CB LCs. **g** Analyzing the effect of printing resolution and 5CB content on the structure of 3D-printed products. The orientation degree was varied by adjusting the printing resolution of PR-5CB-3// from 25 to 100 μm, and the aggregated LCs floated on the surface to form microcrystals after adjusting the printing contents from PR-5CB-3// to PR-5CB-7//. **h** Proposed mechanism: 5CB molecules preferentially align in a direction parallel to the UV light vibration vector, and the orientation of 5CB subsequently governs the oriented polymerization of monomers or prepolymers possessing acrylate groups. All the data in (**a–d**) were collected five times and the error bars represent the standard deviation.

resulting in a milky coloration. When the PR-5CB-3⊥ model was used, the orientation of the 5CB LCs and polymer chains was parallel to visible light. Upon visible light irradiation, the 5CB and polymer chain reflected less light than those in the PR-5CB-3// samples, which allowed the passage of visible light along the orientation of its 5CB LCs and polymer chains, resulting in its transparency. A uniform orientation arrangement of LC molecules obtained in PR-5CB printed samples is a prerequisite for their use in displays and optics. Therefore, complex models were produced using different printing orientations to investigate the applicability of PR-5CB printed samples in optical devices. The innovative PR-5CB 3D printing material requires only simple processing, enables microscopic control of the molecular orientation, and is not limited to thin-film products, significantly reducing the cost of the product. Therefore, it is promising for applications in areas such as LC displays and information recording.

### Investigating the influence of the printing resolution and LC content

Hereinafter, only vertically printed models were assessed because of their improved performance. The LC content associated with the material was another key factor affecting the oriented structure.

Samples printed with various contents of 5CB LCs (3%, 5%, and 7%) were used to study the effect of 5CB content on the orientation state. Herein, 3D printing was used to study the light-oriented photocurable resins. The length of 5CB LCs was roughly between 13 and 20 μm based on the POM images in Supplementary Fig. 1a. Therefore, it was considered that 3D printing serves to enhance the polymer by anchoring the 5CB LCs in each printed layer through the polymerization process of the photosensitive resin into the polymer during the layer printing process. In addition, the change in polymerization that results from varying the printing precision (25, 50, and 100 μm) was assessed to determine the optimal content of added 5CB and printing resolution.

First, the mechanical properties of the 3D-printed samples obtained using PR-5CB resins with various printing resolutions were investigated. The stress–strain and bending strain curves are shown in Supplementary Fig. 3, and the tensile and flexural strengths are shown in Fig. 4a and Fig. 4b, respectively. In addition, the impact strength and hardness are summarized in Fig. 4c and Fig. 4d, respectively. The results showed that the mechanical properties of the products printed using the resin without LCs (PR-5CB-0//) were almost independent of the printed layer thicknesses; for example, tensile strengths of 41.9, 42.1, and 40.4 MPa were obtained for the thicknesses of 25, 50, and 100 μm, respectively. This indicates that

the effect of changes in the printing resolution on the mechanical properties of the PR-5CB-0// products is marginal. However, the addition of 3% of 5CB LCs and maintaining a certain printing resolution can effectively improve the mechanical properties of the 3D-printed products. In particular, the increase in the mechanical properties was highest at a resolution of 25 μm. The tensile strength, elongation at break, flexural strength, flexural modulus, hardness, and impact strength for PB-5CB-3//(25 μm) reached 122.2 MPa, 23.4%, 222.1 MPa, 2836.8 MPa, 75.5HD, and 11.09 kJ/m² respectively. These values were 291.6%, 178.6%, 270.2%, 185.7%, 102.8%, and 275.2% higher than the corresponding values for PB-5CB-0//(25 μm), respectively. Therefore, the strength and toughness of the product were effectively increased by the addition of 3% of 5CB LCs. However, the mechanical properties decreased when the printing resolution was increased to 50 μm and 100 μm. The decrease in the mechanical properties at the printing resolution of 100 μm was so significant that they were even lower than those of the products printed with the PB-5CB-0// resin. The tensile strength, elongation at break, flexural strength, flexural modulus, hardness, and impact strength were 102.1 MPa, 23.4%, 156.5 MPa, 2556.9 MPa, 73.5 HD, and 8.62 kJ/m² for PB-5CB-3//(50 μm) and 32.8 MPa, 15.4%, 104.7 MPa, 2012.9 MPa, 71.6 HD, and 6.22 kJ/m² for PB-5CB-3//(100 μm). In addition, the mechanical properties of the printed product gradually decreased as the LC content increased. For example, the tensile strength, flexural strength, flexural modulus, and impact strength decreased from 122.2 MPa, 222.1 MPa, 2836.8 MPa, and 11.09 kJ/m² for PB-5CB-3//(25 μm) to 109.3 MPa, 106.7 MPa, 2034.1 MPa, and 10.96 kJ/m² for PB-5CB-5//(25 μm) and then to 82.9 MPa, 86.5 MPa, 1763.4 MPa, and 8.79 kJ/m² for PB-5CB-7//(25 μm). This shows that simply adding the 5CB LCs does not necessarily enhance the performance of 3D-printed products; rather, the performance can be effectively enhanced only by controlling the printing resolutions. Mechanical performance metrics for various 3D-printed samples crafted using distinct photosensitive resins are detailed in Supplementary Table 2. This data reveals that commercial photosensitive resins typically exhibit limited tensile strength. When fillers are introduced to augment this tensile strength, the resultant product's toughness invariably diminishes. Contrarily, the PR-5CB resins conceptualized in this research boast superior mechanical properties. Notably, they allow for a concurrent enhancement of both tensile strength and toughness. Evaluating the comprehensive performance of the finished product, PR-5CB-3 emerges as the most commendable resin.

To bolster the assertion that the orientation of the 5CB LCs enhances the toughness of the printed materials, the impact fractures of printed objects with diverse printing resolutions of PR-5CB resins were scrutinized via SEM. The findings are depicted in Supplementary Fig. 4. Examining the PR-5CB-0 system across varying printing resolutions, we noted smooth cross-sections and microcracks that were oriented singularly. This singular orientation stems from the inhibited molecular movement of the cured resin, complicating the manifestation of yield deformation. During their expansion, these microcracks faced minimal resistance, a trait characteristic of brittle failure. The striations observed on the fractured surfaces originated from the energy absorbed during the product's fracturing. Notably, the fracture surfaces of the 25 μm printed samples displayed more pronounced and irregular tearing patterns, accompanied by noticeable creases. This accentuates the remarkable toughness of the printed items. Moreover, as the printed layer thickness escalated, the ruggedness of the fractured surface also intensified, aligning with the previously discussed impact strength results.

The POM image of the surface of the 3D-printed PR-5CB-0// products was dark throughout the 360° stage rotation (Supplementary Fig. 5), indicating that the surface was isotropic at different printing resolutions. For the 25 and 50 μm printing resolutions, the PR-5CB-3// and PR-5CB-5// products showed optical anisotropy similar to that of LCs, indicating that the photosensitive resin polymerizes along the orientation direction of small LCs during the polymerization process. The POM images show that rotating the carrier table by 360° results in an alternation between four bright and four dark images. At the 100 μm printing resolution, the POM images of the PR-5CB-3// and PR-5CB-5// samples did not alternate between light and dark throughout the 360° rotation on the carrier table. However, these samples were always bright, indicating that cryptocrystalline or microcrystalline aggregates were present on the sample surface. Regarding the PR-5CB-7 samples, the POM images were always bright, and some spherical aggregates were observed on the surface even when the printing resolution changed from 25 μm to 100 μm.

XRD can accurately describe the arrangement of LC molecules. For nematic LCs, no diffraction peaks appeared in the small-angle region; however, a dispersion peak appeared in the wide-angle region of 2θ = 20°. The size, height, and width of this peak were related to the content of this phase in the sample, degree of dispersion, and orderliness. Generally, the higher and larger the peak, the more (orderly) surface phases are present. Therefore, the XRD of the thin surface section of the 3D-printed PR-5CB-0// and PR-5CB-3// samples at 25, 50, and 100 μm printing resolutions was tested to verify the orderly variation of LCs (Fig. 4e). The results show that there was little variation in the peaks of the PR-5CB-0-printed products at different printing resolutions. However, the XRD peak of the PR-5CB-3 sample at a printing resolution of 25 μm was higher than that of the PR-5CB-0 sample. The thickness of the PR-5CB-3 sample was approximately the same as the length of the rod LCs and the printing time was short. Therefore, the LCs oriented the surrounding photosensitive resin in the same direction as that of light irradiation. When the photosensitive resin polymerized, it fixed the 5CB LCs in one direction, resulting in the polymer exhibiting anisotropy. Increasing the printing resolution to 50 μm resulted in a marginal decrease in the XRD peak. Because the thickness of the polymer was greater than the length of the LC rod, the LCs also aligned with the surrounding photosensitive resin during light orientation. However, the orientation was incomplete when the quantity of photosensitive resin was too high. When the printing accuracy was increased to 100 μm, the height of the XRD peak significantly decreased, and the 5CB LCs were surrounded by the photosensitive resin. Consequently, the photosensitive resin was dispersed as it could not be oriented during light treatment. This resulted in a relatively random orientation that the polymer could not adopt. The effect of different contents of 5CB on the polymer orientation during 3D printing was examined. The XRD curves of PR-5CB-3//(25 μm), PR-5CB-5//(25 μm), and PR-5CB-7//(25 μm) were compared, and the height of the XRD peaks decreased as the 5CB content increased. In particular, when the 5CB content reached 7%, the height of the XRD peak sharply decreased. This implies that an increase in 5CB content results in LC aggregation, making it more difficult for the LCs and the photosensitive resin to orient with light irradiation.

To demonstrate that polymer orientation also occurs, two methods were used: first, dichloromethane (DCM) was used to remove the small 5CB LCs from the PR-5CB-3//(25 μm) samples while leaving the polymer behind; the surface of the product was examined using the POM. The phenomenon of alternating four bright and four dark images was observed when rotated by 360° using the carrier table (Supplementary Movie 7). The bright and dark images are shown in Fig. 4f. This shows that the photosensitive resin also undergoes orientation during the polymerization process, forming a structure with optical anisotropy similar to that of LCs. In the second method, the PR-5CB-3//(25 μm) sample was observed using POM while being warmed at 30, 60, 90, 120, and 150 °C (Supplementary Fig. 6). By observing the POM images of PR-5CB at 10 × 40 magnification, a clear change from completely dark to completely bright during the rotation of 0°–45° between cross-polarizers was observed. This change in characteristics proves that the polymer chain segments are anisotropic, indicating

that some orientation of the polymer chains occurs. This is because thermotropic LCs (5CB), which have a unique protruding rod-like structure, reversibly change the spatial orientation of their bodies as the local temperature of the molecule is adjusted by the increase in temperature. LCs can transition from the randomly oriented state of the isotropic phase to the aligned state of the isotropic phase when below the isotropic transition temperature (30 °C). In addition, their POM maps appear to alternate between light and dark in a 360° rotation between cross-polarizers. When the temperature of the LCs is raised above the isotropic transition temperature, 5CBs lose orientation order, resulting in POM diagrams that are always dark. However, the POM images of the polymer are always observed to be distinctly light and dark, even when 5CB loses its oriented order as the temperature rises. This distinctive feature of the change is made further evident by the extent of the orientation of polymer chain segments.

The same two methods can also be used to verify that the spherical LCs were immobilized on the surface of the samples and transformed into microcrystalline structures after cooling. First, the surface film of PR-5CB-7//(50 μm) was soaked in dichloromethane for 6 h to remove the 5CB LCs; subsequently, it was subjected to POM observation (Supplementary Fig. 7). The surface film of PR-5CB-7 (50 μm) with 5CB present is shown in Supplementary Fig. 7a; this film has aggregated spherical LCs on its surface. However, the aggregated spherical LCs were not observed on the surface film of PR-5CB-7 (50 μm) with 5CB absent (Supplementary Fig. 7b). Because of an increase in LC content, the anchor point provided by the photosensitive resin was not able to induce partial LC orientation, resulting in the retention of 5CB in the form of droplets. Regarding the 3D-printed PR-5CB-7 sample, the lower surface orientation and LC aggregation resulted in a significant decrease in tensile strength and modulus. The droplet form of 5CB was not conducive to the improvement of product strength. In the second method, the surface film of PR-5CB-7 (50 μm) was selected and placed on a heating table for concomitant heating and POM imaging, and the results are shown in Supplementary Fig. 7c. As the temperature increased, the spherical crystal aggregates on the surface of the film gradually disappear until they become completely black. This indicates that the spherical crystals slowly change from the crystalline state to the disordered state. This further verifies the aggregation of 5CB during the printing process when it is added in surplus to the resin.

The diagram of the mechanism in Fig. 4g is used to explain and summarize the experimental results above. First, the effect of printing resolution on 5CB and polymer orientation was explored. In the printing process, a certain laser power is required for the photoelectric field to change the molecular orientation, which gives rise to the orientation of the 5CB LC molecules. At a printing accuracy of 25 μm, the thickness of the printed layer is approximately the same as the length of the rod-like LCs, and the printing time is short; therefore, the LCs orient the surrounding photosensitive resin with the direction of light irradiation. Subsequently, when the photosensitive resin polymerizes, it anchors the 5CB in one direction, giving rise to the anisotropic character of the polymer. This indicates that the photosensitive resin also undergoes orientation during the polymerization process, forming a structure with optical anisotropy similar to that of LCs. When the printing resolution was increased to 50 μm, the LCs were oriented with the surrounding photosensitive resin in the direction of light irradiation because the thickness of the polymer layer was greater than the length of the LC rods. However, because too much photosensitive resin was present, its orientation was incomplete. When the printing resolution was increased to 100 μm, the 5CB LCs were surrounded by the photosensitive resin, restricting its ability to orient and disperse during illumination. Therefore, the orientation of the resin was relatively random, preventing the polymer from following the orientation of the LCs. Second, the effect of the variation in 5CB content on

polymer orientation was explained. The degree of LC orientation of PR-5CB-3// was strong; however, the anchoring point provided by the photosensitive resin could not induce partial LC orientation because of an increase in LC content. This resulted in the retention of 5CB in the droplet form. For the PR-5CB-7// 3D-printed samples, the lower orientation and the presence of LC aggregates on the surface resulted in a tensile strength and modulus that were significantly lower than those of the PR-5CB-3// sample. In addition, the droplet form of 5CB did not favor improvement of the product strength. Because of the long printing time and high polymerization heat, the 5CB LCs were randomized and agglomerated into spheres above the orientation temperature; the aggregated LCs floated on the surface of the printed sample. However, after the photosensitive resin was converted to the polymer, these spherical LCs were fixed on the surface of the sample, forming microcrystalline structures upon cooling.

To further demonstrate that the strategy of light-oriented 3D printing of 5CB LC/photocurable resins is effective across all acrylic photocurable resins, the following tests were performed. First, different commercial printing formulations (Supplementary Table 3) with 3% 5CB content were selected; sample strips of these formulations were printed at 25 μm printing resolution to test their tensile properties (Supplementary Fig. 8 and Supplementary Table 4) and examine their POM images (Supplementary Fig. 9). Alternating four bright and four dark images were observed for all printing formulations when rotated by 360° using the carrier table (Supplementary Movies 8–10). These results demonstrate that all photosensitive resins with added 5CB are well-oriented when exposed to light, which improves their mechanical properties. Second, experiments were designed to investigate the role of various prepolymers in the light orientation of 5CB samples by designing multiple formulations, preparing various types of photosensitive resins containing 3% content of 5CB (see Supplementary Table 5 for the formulations), and conducting POM tests on the printed samples. The results are shown in Supplementary Fig. 10. Alternating four bright and four dark images were also observed for all formulations, indicating that they were all oriented by the behavior of 5CB. This verifies that if the modified resin has acrylate photosensitive groups, it can be oriented by the behavior of 5CB. Figure 4h illustrates the hypothesized orientation mechanism wherein 5CB molecules exhibit a preference to align in a direction parallel to the UV light vibration vector. This orientation of 5CB subsequently governs the oriented polymerization of monomers or prepolymers that possess acrylate groups.

## Analyzing the thermal stability and 3D printing accuracy of the PR-5CB printing samples

Dynamic mechanical thermal analysis (DMA) is an important tool for the characterization of the compatibility of photosensitive resins with 5CB LCs and the curing and crosslinking of polymers. Therefore, the temperature variations in the energy storage modulus and loss factor of the products printed using PR-5CB resins at 25 μm printing resolution in the printing direction parallel to the stretching direction were further investigated using the DMA technique, and the results are shown in Fig. 5a, b. The introduction of the 5CB LCs significantly improved the energy storage modulus of the product. The storage modulus values for PR-5CB-3//(25 μm), PR-5CB-5//(25 μm), and PR-5CB-7//(25 μm) were 1702.3, 1629.8, and 1555.6 MPa, respectively. These values were 194.4%, 186.2%, and 177.7% higher than the respective value for PR-5CB-0//(25 μm) (875.4 MPa). The $T_g$ values for PR-5CB-3//(25 μm), PR-5CB-5//(25 μm), and PR-5CB-7//(25 μm) were 91.6, 89.2, and 88.4 °C, respectively; all were higher than that of PR-5CB-0//(25 μm) (71.7 °C). However, the gradual decrease in the energy storage modulus is particularly noteworthy because it indicates that the stiffness of the printed product decreases as the 5CB content increases. This result is attributed to the reduction in the crosslinkage of the polymer. The

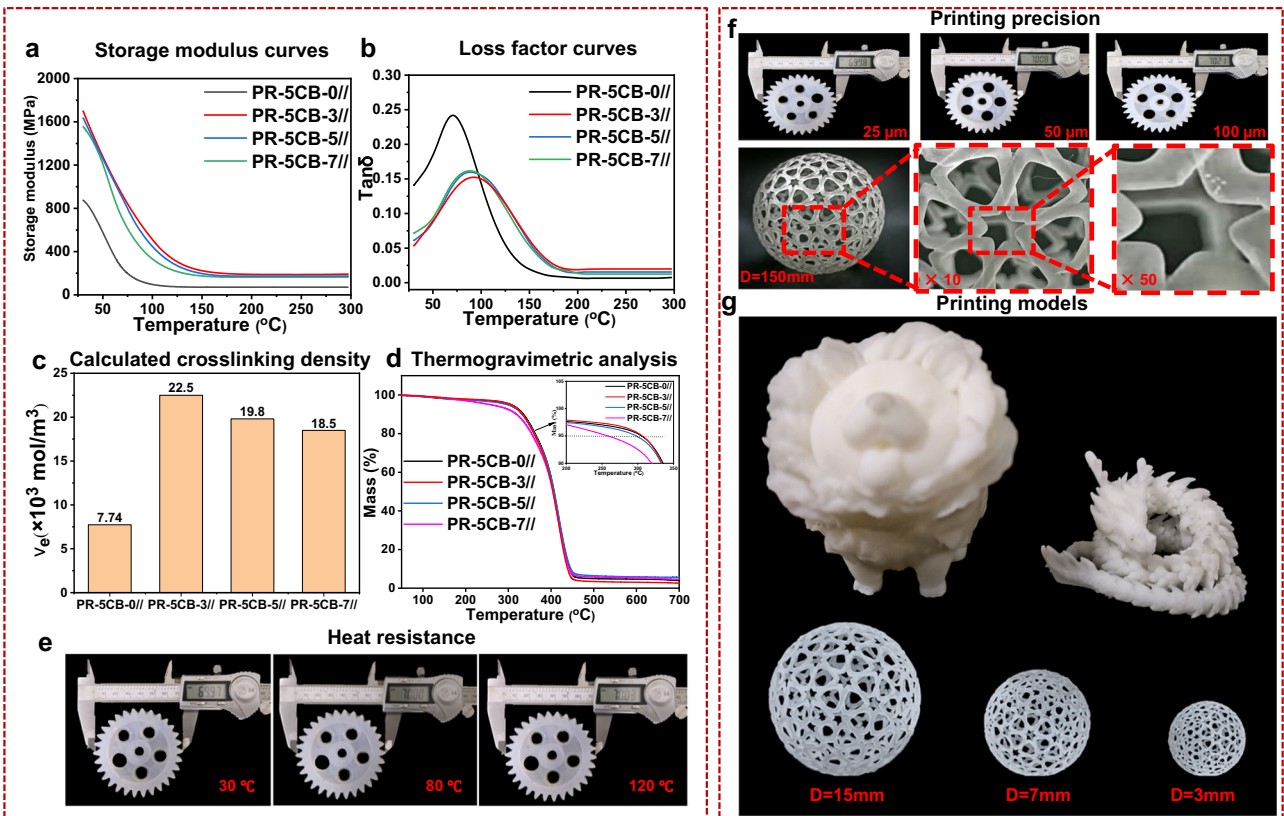

**Fig. 5 | Thermal stability and 3D printing accuracy of the PR-5CB printing samples. a** Storage modulus curves. **b** Loss factor curves. **c** Calculated crosslinking density. **d** TGA curves. **e** Printing model of PR-5CB-3//(25 μm) heat-treated at different temperatures. **f** Shrinkage of PR-5CB-3// resin at different printing resolutions and skeletonized spheres with a resolution of 2833 × 2918 pixels and height of 150 mm magnified by 10× and 50×. **g** Printing model of macro- and micro-sized structures.

crosslink density is generally calculated using the theory of elastic dynamics of rubber:

$$Ve = E'/3RT,$$

where $E'$ is the storage modulus in the rubbery state ($E'$ at $T_g + 50\,°C$), $R$ is the gas constant (8.314 J/mol·K), and $T$ is the absolute temperature at $T_g + 50\,°C$. According to the calculated results (Fig. 5c), the crosslinking density of PR-5CB-3//(25 μm) was the highest at $22.5 × 10^3$ mol/m³ and decreased with increasing LC content. Therefore, an addition of more than 3% of 5CB will reduce the crosslinking density of the system from $22.5 × 10^3$ to $18.5 × 10^3$ mol/m³, reducing the energy storage modulus of the resin. The difference in the crosslinking density between PR-5CB-3//(25 μm) and PR-5CB-0//(25 μm) was due to an increase in physical crosslinks, the presence of rigid structures in 5CB, and the anchoring force between the well-ordered LCs and the polymer, making the system more compact. When the 5CB LC content further increases, the unoriented LCs agglomerate and weaken the molecular network interactions. The thermal stability performance of the printed samples was demonstrated by the thermogravimetric analysis (TGA) curves (Fig. 5d). The thermal decomposition curves based on PR-5CB resins show similar trends, with no significant differences in the weight loss rates, and are mainly divided into three weight loss stages. For example, the initial ($T_{-5\%}$) and maximum ($T_{max}$) decomposition temperatures of the PR-5CB-3//(25 μm)−312.0 and 423.5 °C, respectively−were approximately 7 and 5 °C higher than those of PR-5CB-0//(25 μm) (306.3 °C, 418.4 °C), indicating that addition of the appropriate amount of 5CB enhances the thermal stability of the printed samples. This result can be explained by two effects: (1) the rigid benzene ring structure in 5CB restricts the motion of the chain segments in the system; (2) the short

rod-like LC structure accounts for a small fraction of degrees of freedom in the system and can be tightly interspersed in the molecular network at high temperatures. Further increasing the 5CB content marginally decreases the $T_{-5\%}$ and $T_{max}$ values to 296.2 and 413.1 °C, respectively. Under these conditions, 5CB largely exists in droplet form, reducing the degree of crosslinking between the polymer molecules and facilitating the movement of the molecular chain segments. To ensure that the printed models remained intact post high-temperature exposure, PR-5CB-3//(25 μm) printed samples underwent heat treatment at various temperatures for a duration of 24 h. Despite the escalation in temperature, the printed items exhibited no deformation. Furthermore, any shrinkage in the samples at elevated temperatures was marginal, remaining under 0.05%, as presented in Fig. 5e.

Figure 5f shows the gear model of the PR-5CB-3// resin at different printing resolutions. The diameter of the gear stl. file of 70.00 mm and the formulated PR-5CB-3// resin with the best mechanical properties for printing was chosen. The diameter of the PR-5CB-3//(25 μm) resin printout was 69.98 mm with a relative error of 0.03%; that of the PR-5CB-3//(50 μm) resin was 70.08 mm with a relative error of 0.12%; and that of the PR-5CB-3//(100 μm) resin was 70.21 mm with a relative error of 0.30%. The skeletonized sphere remained structurally intact as observed at different magnifications, indicating that high quality can be achieved. Figure 5g shows that models with the required precision were printed using the PR-5CB-3 resin. The details of each model are clearly visible, illustrating the potential use of this photosensitive resin in producing advanced precision instruments. Micro-sized structures were also obtained, and the minimum skeletonized sphere size that could be printed is 3 mm in diameter. Overall, these results demonstrate that the PR-5CB resin prepared in this study has high printing resolution and can be used for printing various products with high accuracy requirements.

## Discussion

In this study, PR-5CB resins were successfully prepared by blending 4'-pentyl-4-cyanobiphenyl with an acrylic photosensitive resin; subsequently, SLA 3D printing was used to print products using this resin. The rheological properties of PR-5CB resins provided fast light-curing abilities. By controlling the 3D printing method, printing resolution, and 5CB content, the mechanical properties of 3D-printed products could be effectively improved. The light-driven orientation process of 5CB guides other acrylate prepolymers to orient in the same direction; therefore, the entire fixed 3D printing polymer was observed to be anisotropic by POM. The tensile strength, elongation at break, flexural strength, flexural modulus, and impact strength of PR-5CB-3//(25 μm) were 122.2 MPa, 23.4%, 222.1 MPa, 2836.8 MPa, and 11.09 kJ/m$^2$, respectively. These values are 2.91-, 1.80-, 2.70-, 2.12-, and 2.75-fold higher than the corresponding values of commercial inks. Printed products based on the PR-5CB-3//(25 μm) resin have excellent surface quality, less than 0.05% shrinkage, and good heat resistance; additionally, the minimum skeletonized sphere size that could be printed was 3 mm in diameter. The method introduced in this study is simple and inexpensive and represents an innovative solution for 3D-printing LC-assisted light-driven orientation. The excellent mechanical properties of the obtained printed products can broaden the application range of AM to high-end fine displays and optical devices. The use of 3D printing technology for the fabrication of device systems is an approach that requires further exploration, and the use of PR-5CB resins for 3D printing is significant for the development of self-reinforced composite materials that meet the needs of different fields.

## Methods

### Materials

Hydroxyethyl methacrylate (HEMA) (≥99%) and 2,4,6-trimethylbenzoyl diphenylphosphine oxide (TPO) (≥97%) were obtained from Rohn Chemical Reagents (Shanghai, China). Aliphatic urethane acrylate (CN9010) (≥99%), polyurethane acrylate (CN991) (≥99%), and ethoxylated pentaerythritol tetraacrylate (SR494) (≥99%) were purchased from Juncai Material Technology Co., Ltd. (Shanghai, China). 4'-Pentyl-4-cyanobiphenyl (5CB) (≥99%) was purchased from Shanghai Biogenic Leaf Co., Ltd. (Shanghai, China). All the materials were used as received without further purification.

### Preparation of PR-5CB resins

Photosensitive resins containing different quantities of 5CB were prepared using solution blending. The PR-5CB resins were prepared by mixing different amounts of CN9010, CN991, SR494, active diluent HEMA, and photoinitiator TPO. Mixtures were emulsified and stirred at 80 °C for 30 min and then cooled to room temperature. Finally, a specified quantity of 5CB was added into the mixture. These resins were labeled according to their compositions as described in Supplementary Table 1.

This resin facilitates the evaluation of how LC additives influence the efficacy of photosensitive resins. Moreover, it paves the way for ensuing mechanistic investigations aimed at augmenting the performance of these resins. Concerning the proportion of 5CB incorporated, the experimental findings refined the requisite quantity to a specific range. As a result, in this research, 5CB was added to LCs at concentrations of 3%, 5%, and 7%. Introducing even a modest percentage of 5CB can markedly bolster the resin's mechanical attributes, thereby ensuring cost-effective production while enhancing its practical utility.

### Stereolithography (SLA) printing

The 3D molded components were fabricated using an SLA printer (Form 2, Formlabs Inc., Somerville, MA, USA), which was equipped with a 405 nm wavelength laser light source and characterized by a laser power and spot size of 250 mW and 140 μm, respectively. The derived laser intensity can achieve 10$^5$ W/cm$^2$, a prerequisite for orienting the chosen 5CB LCs. The model was first designed using the SolidWorks software, followed by layering using the printing software to control the UV light-curing time and the exposure state of the liquid resin; this allowed the photographic resin to be printed layer-by-layer according to the pre-designed modeling structure. The printing strategy focused on printing molded parts by controlling the thickness of the print monolayer. The obtained samples were washed with ethanol and dried at 30 °C under airflow.

Tensile, bending, and notched specimens were printed using a Form 2 3D printer according to the Chinese GB/T 1040.1-2018, GB/T 9341-2008, and GB/T 18658-2018 national testing standards, respectively. These samples were post-treated with UV light for 10 min and left to stand for 24 h prior to measurements and performance tests.

### Extraction 5CB from 3D-printed products

DCM was used to remove the small LCs (5CB) from the PR-5CB-printed product. After many hours of immersion, the 5CB was removed by DCM extraction until the mass of the sample remained constant.

### Polarizing optical microscopy

A POM (59XA-2) was used to observe the dispersion state of the photosensitive resins and the orientation of the 3D-printed films. An appropriate amount of dispersed liquid was dropped between two cover glasses to form a thick layer of solution, and its state was observed. Samples for observing the surface and cross-section of the printed samples were obtained by laying them flat on a table and using a cutter to intercept the upper surface and cross-sectional area of the film, which had a thickness of 1 mm. Moreover, the film was placed between two coverslips for observation using a polarized light microscope. Subsequently, the circular platform of the microscope was rotated, and light and dark changes in the shooting field of vision were observed.

### Rheological behavior measurement

The rheological behavior of the resin was measured using a TA Instruments Discovery Hybrid Rheometer (DHR-2). The steady-state shear rates ranged from 1 to 1000 s$^{-1}$. The kinetics of the light-curing process were evaluated using a rheometer equipped with a UV light-emitting diode (LED). The gap between the two geometries was set at 0.1 mm. The upper and lower plates comprised aluminum and transparent polymethyl methacrylate, respectively. The duration of the experiment was 80 s, and the samples were irradiated at an intensity of 80 mW/cm$^2$.

### X-ray diffraction test

A 3D-printed product's degree of orientation was examined using an Ultima III X-ray diffractometer (Japan). The equipment featured a copper target, and the set parameters for the examination included a tube voltage of 40 kV, tube current of 40 mA, scanning range from 5–80°, and a scanning speed of 10°/min. Samples measuring 12 mm × 12 mm × 12 mm were printed using a 3D printer, with adjustments made to both the printing direction and resolution. These samples were subsequently bisected into two identical 6 mm × 6 mm × 6 mm squares. During the evaluation of the surface and cross-sections of specimens from various printing methodologies, the printing surface and cross-section were oriented towards the top. From these, samples with a 1 mm thickness were carefully sliced using a knife. The specimens were then uniformly sanded using sandpaper, ensuring a level and polished test surface.

### Tensile test

The tensile properties were evaluated using a universal material testing machine (LD24, Labsans, China). The tensile test speed was 10 mm min$^{-1}$. The test results reported for each sample are the average values of five replicate tests.

## Notched impact performance

A WH-8050 digital pendulum impact tester (Ningbo Zhenhai Weiheng Testing Instrument Co., Ltd.) was used for the notched impact performance test. According to the GB/T 1043-2008 standard, the test sample dimensions were $80 \pm 2$, $10 \pm 0.2$, and $4 \pm 0.2$ mm, and the V-shaped notch depth was $2 \pm 0.1$ mm. The test results reported for each sample are the average values of five replicate tests.

## Hardness

Hardness was tested using a D-type digital rubber hardness tester according to the GB/T 531-99, GB/T 2411-80, HG/T 2489-93, and JJG 304-003 standards. The test results reported for each sample are the average values of five replicate tests.

## Dynamic mechanical analysis

Dynamic mechanical analysis (DMA) was performed using a DMA Q800 instrument from TA Instruments (USA). DMA tests were performed in the double cantilever in the temperature range of 30–300 °C at a heating rate of 3 °C/min and frequency of 1 Hz, using a sample with the dimensions of 35.0 mm × 10.0 mm × 3.0 mm.

## Scanning electron microscopy

SEM (JEOL JEM-2010, Japan) was used to observe the fracture surface morphology of the 3D-printed parts. The gold film was sputtered on the sample surface with a magnetron sputterer, and the scanning acceleration voltage was 10 kV.

## Thermogravimetric analysis

TGA was performed using a STA449C instrument from TA Instruments (USA) at a temperature range of 30–800 °C and a weight of 5–10 mg under a nitrogen atmosphere with a flow rate of 100 mL min$^{-1}$ and a ramp rate of 10 °C/min$^{-1}$.

## Reporting summary

Further information on research design is available in the Nature Portfolio Reporting Summary linked to this article.

## Data availability

The authors affirm that the data buttressing the conclusions of this research can be found both within the manuscript and its Supplementary Information files. If there is a need for raw data in alternative formats, they can be procured from the lead author upon request.

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

## Acknowledgements

This research was financially supported by the National Natural Science Foundation of China (52103356, 51773038), Fujian Provincial Department of Science and Technology (2019Y0042, 2019J01730, 2020H0045, 2020J01770, 2020J01773, 2020J01150, 2021J01963), Regional Development Projects of Fujian Province (2020H4017), Major Special Project of Fujian Provincial Department of Science and Technology (2021HZ027003), Program for Innovative Research Team in Science and Technology in Fujian Province University (IRTSTFJ), Fujian Provincial Central Guidance for Local Science and Technology Development Funds Projects (2022L3026), Major Science and Technology Projects of Quanzhou (2021GZ2, 2022GZ5, 2023GZ1), the Bureau of Science and Technology of Quanzhou (2020C060), and the Fund of Fujian Innovation Center of Additive Manufacturing (ZCZZ202-33), and Student Innovation and Entrepreneurship Training Program of Quanzhou Normal University (202310399008).

## Author contributions

S.C., Q.C., and D.Z. conceived and designed the research; X.S., S.C., B.Q., and R.W. performed the experiments; X.S., S.C., Y.Z., X.L., W.L., and J.G. analyzed the data; S.C. and X.S. wrote the paper; All the authors discussed the results and commented on the manuscript. S.C., Q.C., and D.Z. supervised the study.

## Competing interests

The authors declare no competing interests.
