## [Peer Review File · Nature Communications]

Light-oriented 3D printing of liquid crystal/photocurable resins and in-situ enhancement of mechanical performanceREVIEWER COMMENTS

Reviewer #1 (Remarks to the Author):

In this manuscript, Zhuo, Chen, Chen, and co-workers describe how a commercially relevant liquid crystal small molecule (5CB) can be added to an industrial resin to improve its thermomechanical properties, while additionally providing unique optical anisotropy to the printed parts. The authors perform a thorough examination of the 5CB containing resin dispersion via rheology, printing parameters and their effects on optical, thermal, and an array of mechanical properties, including tensile strength, elongation and break, flexural strength, impact strength, and storage strength. To the latter point, the authors show distinct improvements in mechanical properties for the composite structures relative to the analogous parts without 5CB present. While these results appear significant and of general interest to the readership of Nature Communications the manuscript was at many times quite difficult to follow, particularly as it relates to the mechanism of orientation.

Furthermore, a rationale for the system selected and details on composition in the main text were lacking making it appear at the outset to be somewhat random in-terms of how the particular resin system was selected (apart from the use of 5CB, which was described in the introduction). Finally, a number of claims made by the authors are not adequately supported by the evidence provided, such as the mechanism of LC alignment, the purported alignment of polymer chains, and the rationale for enhanced mechanical properties. Thus, while the results are interesting and have potential to be of significance, the manuscript is not recommended for publication at this time. Below are a number of comments/questions to the authors provided in order of the manuscript that may be able to assist with clarity for publication upon major revisions.

Comments/questions:

- 1) Minor correction – SLA stands simply for “stereolithography” not “stereo lithography appearance” as written in both the abstract and introduction.
- 2) The authors are encouraged to remove the exaggerated statement of “unlimited potential” in the last sentence of the abstract.
- 3) The direct reference to “CLIP” at the end of the first paragraph is somewhat misleading since it is not used by the authors. Instead, the authors are encouraged to incorporate this

reference into a previous sentence (line 48 perhaps) to include a statement on the high manufacturing speed of SLA, without directly writing CLIP so-as to not mislead the reader.

4) I find the second paragraph of the introduction on line 53 of page 2 to be inaccurate and contradictory to statements made by the authors in the previous paragraph. As such, I strongly encourage that the authors remove a number of statements here that are an attack on objects produced by SLA (lines 53-61), and to refocus on the opportunities to improve material properties beyond what has been achieved so far, which the authors begin to do on line 62 "Several studies...". In actuality, the breadth in mechanical properties of parts produced by SLA is quite good (see for example: <https://bomar-chem.com/markets/3d-printing> and <https://www.carbon3d.com/materials>), and what this article showcases is that LC additives represent a unique avenue to further improving these properties.

5) The introduction provides alignment mechanisms starting at the top of page 4 (line 86), but it was very difficult to follow. It was unclear what the driving force for thermal alignment would be for both the 5CB and polymer chains – is there precedent for polymer chain alignment in standard SLA processes? The authors then go on to describe the photoelectric field effect for molecular orientation, which requires 10^2 - 10^3 W/cm² light intensities, which is possible for laser-based SLA, but not projection-based processes capable of providing the fast speeds mentioned by the authors in the introduction (see comment 3). However, the type of SLA 3D printer was not explicitly specified (it is a Form 2 from Formlabs, which is a laser-based system – but that should be stated) nor was the light intensity used in this system specified – and that should also be included along with the spot size of this printer. On line 120 the authors then comment on the lamination direction, eluding to the importance of this in the liquid crystal orientation. Thus, it is very confusing after reading the introduction to know what effect or effects may be contributing to the reported LC alignment in this process. As such, the authors are encouraged to better clarify this text and provide a schematic to this point to assist readers in understanding the important mechanisms at play.

6) At the beginning of the results and discussion section the authors are encouraged to clearly state what commercial resin was selected for this study and why that one in particular was selected. Additionally, the authors should state how much 5CB is added to the resin and why that amount was chosen at this stage (it only becomes apparent much later).

- 7) The authors note the intensity of irradiation for the rheology to be “80 MW/cm²”, however this is likely a typo and should be “mW/cm²” – the authors should clarify this.
- 8) The authors claim that the presence of the 5CB in the resin increases the curing speed of the resin, but the difference shown does not appear statistically significant, nor do the authors provide standard deviations for this claim and thus it should be removed.
- 9) Similar to the above point, the authors are strongly encouraged to provide +/- 1 standard deviation from the mean of each measurement reported. According to the experimental section >3 samples were used for each mechanical measurement and as such this data should be available and will provide significance to the reported results. This is important as mechanical properties can be quite defect-dependent, leading to large variability.
- 10) The claim that LC alignment along the stacking direction improves mechanical properties based on the results shown Figure 2B does not seem substantiated to me as the green and blue curves appear quite similar, and all four show somewhat unusual stress-strain responses for a rigid plastic, where there is an initial low Young’s modulus followed by a sharp increase in stiffness at different strains ~5-7%. The authors should comment on this unusual behavior. Furthermore, the difference in mechanical properties follow what is typical for printing horizontal vs vertical layers, where the larger difference observed for the 5CB cases could simply be ascribed to the increased heterogeneity, as observed in the optical data of Figure 3, leading to worsened layer-to-layer bonding. Thus, more evidence to support this claim surrounding the effects of molecular anisotropy on mechanical anisotropy should be provided.
- 11) The plots in figures 2 and 5 were very difficult to read. The authors are encouraged to increase the size of the plots and associated text to make them more legible.
- 12) The paragraph starting on page 8 (line 207) describing the mechanism of fracture mechanics for brittle failure of highly crosslinked thermosets is unnecessary to include in the main text as it does not add insight for the specific system describe, but rather follows standard behavior for such materials. The authors are encouraged to remove this paragraph or put it in the supporting information file.
- 13) Paragraph 9 refers to the improved “toughness” of the 5CB containing samples, but “toughness” as it is being referred to here is not defined. The authors should define this and reference the associated data that corroborates the statements made about toughness enhancements in the presence of 5CB.

14) The SEM images in Figure 3 clearly show a difference between samples with and without 5CB, but it is not clear from these images why the striations are present and what direction the different samples are being viewed at. These points should be clarified.

15) Line 244 references “Fig. 2”, but it should be “Fig. 4”.

16) The authors claim that the polymer chains are aligned based on the results of Figure 4 because it goes from fully light to dark upon rotating 45 degrees between crossed polarizers, as opposed to only seeing “bright lines” (line 256). However, the magnification of the images shown is not provided and proper controls are missing with respect to POM images at the same magnification within the resin at different temperatures. If 5CB is dispersed homogeneously during the printing process then it seems expected to observe uniform changes in brightness under crossed polarizers.

17) Line 280 – the authors state that “the length of 5CB LCs was found to be roughly between 13 and 20 μm based on the POM images discussed above”, but do not reference which images they are talking about nor how those values were determined.

18) Line 302 – the authors refer to an increase in both strength and toughness, but again do not define toughness. If they are referring to the strain energy density (area under stress-strain curve) then this is not a true measure of toughness, which refers to the amount of energy required to propagate a crack, and thus requires an initial defect to be present during measurement.

19) Figure 5 provides error bars for part (d), but they do not appear on the other bar graphs. The authors are encouraged to clarify what these error bars represent (e.g., are they ± 1 standard deviation?) and to include them on the other bar graphs to provide statistical significance.

20) The authors claim that polymer chains are oriented based on the fact that dichroism is observed after washing the sample with dichloromethane. However, the authors do not provide evidence that this washing actually removes the 5CB. If it does, then this would raise concerns about leaching of 5CB over time and a change in properties. This is also contradictory to the statement made by the authors in the introduction (lines 71-72) that the “LCs within the polymer are tightly and stably enclosed.”

21) Figure 6 provides a schematic hypothesis as to changes in alignment for prints made with different thicknesses, but the evidence for this is solely dichroism observed under POM. Additional evidence, such as x-ray scattering would help to substantiate these claims.

Also, the authors should detail what printing direction is being represented in Figure 6.

22) The mechanism described on page 16 with respect to Figure 6b is very detailed, but no evidence for the formation of “spherical liquid crystals...fixed on the surface” is provided to support this text.

23) With respect to the resin selection made above (point 6), the authors claim that the effect of adding 5CB to resins is general and then reference results in the supporting information, specifically noting in the text that optical anisotropy is observed for three additional resin systems (line 414 in the main manuscript and Fig. S2 in the SI). However, the authors fail to comment on the fact that mechanical properties for all three other examples appear unaffected by the incorporate of the 5CB (Fig. S1), which is in contrast to the one shown in the main manuscript. This begs the question as to why this resin is particularly special and needs to be discussed by the authors as it goes against the generality of this approach.

24) Figure 10a shows differences in haze across a 3D object that was uniformly printed either parallel or perpendicular. The authors are encouraged to comment on where this observed contrast arises from when looking at what appears to be planar (flat) objects as in part (a).

25) Line 489 – the authors claim that the printing accuracy of the teeth was “0.14-0.28%”, but it is not clear where this value comes from and how it was measured. The authors are encouraged to expand upon this statement.

Reviewer #2 (Remarks to the Author):

The author prepared a mixture of LC and photosensitive resin. It is interesting to study the arrangement state of LC sandwiched in the tiny gap. A large number of experiments were designed to verify the arrangement state and the improvement of mechanical properties of LC. However, the shortcomings of the article are also obvious. I am not convinced the level of novelty is up to par with the papers often found in Nature Communications and feel it would fit better in NPG Asia Material. I have a few comments:

1. The language needs to be polished; the writing is colloquial and the words are not rigorous and academic; some pictures are not clear; scale bar is missing, too many typos,

and the caption is not specific enough.

2. The arrangement of LC in the printed layer in Figures 4 and 6 should be introduced in Figure 2, which is crucial to help readers understand the difference in mechanical properties. In addition, why are LC molecules aligned vertically to the upper and lower surfaces of the gap (Fig. 6b)? Normally, the long axes of LC molecules tend to be aligned parallel to the upper and lower surfaces of the gap rather than vertically. More scientific explanation is needed.

3. Line135: "a large number of rod-shaped LC molecules", is it a single molecule or grouped molecules? Please clarify.

4. Line342: "extracting 5CB LC", the authors didn't mention any extraction procedure in the Methods section. Please add it if it is necessary.

5. Line393: If the red arrow rotates, then the picture should not be rotated. Why is the POM picture of PR-5CB-0 bright at when the thickness is 50 microns?

6. Line470: "When the printing direction is parallel to the layer stacking direction, the liquid crystal orientation direction is perpendicular to the printing direction, and the model shows a higher haze with a milky white color." Isn't the printing direction always perpendicular to the printing layer? Please clarify.

7. Line519: "Aliphatic urethane acrylate (CN9010), polyurethane acrylate (CN991) ethoxylated pentaerythritol tetraacrylate (SR494)", why are four polymers needed for the fabrication? Please clarify their functions in the light orientation of LCs?

8. LC is a very good optical material. It is not worth to study the improvement of mechanical properties by aligning LC since it is quite expensive compared with other fillers which can reach higher mechanical enhancement. More innovative application related to programmable control of LC arrangement should be added. The redundant mechanical tests should be placed in the support information.

9. Sentences are difficult to understand or have inappropriate terminology, some of them are listed below:

(1) Line21: "the light-driven orientation of 5CB drives other acrylate prepolymers to orient along the orientation direction of 5CB", hard to understand.

(2) Line24: What do you mean by "3D printing lamination method"?

(3) Line29: Too many similar repeated expressions, like "

(4) 121.2 MPa, 25.5%, 222.0 MPa, 11.09 kJ/m², and 1702.3 Mpa respectively; these values

are 281%, 241%, 270%, 275%, and 186% of those of the commercial inks". They need a different expression.

(5) Line38: "molding method and diversified molding", the "molding" is not suitable to describe 3D printing.

(6) Line 57: "the formed parts prepared by the development of the photosensitive resin", hard to understand.

(7) Line175: "PC-5CB-0 \perp ", typo of repeating

(8) Line222: "Fig. 2.", typo

(9) Line237: What is the specific position of the picture in the sample in Figure 3? Need a schematic diagram.

(10) Line274/337: Is it suitable to use "lamination", "printing accuracy" in this article?

Reviewer #3 (Remarks to the Author):

The manuscript presents a printing resin by mixing 4'-pentyl-4-cyanobiphenyl (5CB) with acrylic photosensitive resin. The rheological properties, mechanical properties, and heat resistance of the proposed resins are systematically investigated. Here, there are several concerned questions listed as follows:

(1) Why is 4'-Pentyl-4-cyanobiphenyl (5CB) selected instead of other commonly LCs?

(2) The orientation of LCs is the core for the proposed resin. The advantage should be emphasized both in "introduction" and "application display" again. At present, the printed sample, in manuscript, is too simple.

(3) Will the orientation of LC molecules trigger the contraction of printed sample if temperature exceeds phase transition temperature like previous studies (such as ACS applied materials & interfaces 13(2020): 12698-12708; Advanced Materials 33(2021): 2002541; Science advances 6(2020): eabc0034.)? If the printed sample generates evidently deformation at high temperature (i.e., exceed phase transition temperature), the application of it will be restricted.

(4) The manuscript exaggerates the light-curing ability of the PR-5CB resins. From figure 1c, the light-curing speed of the PR-5CB-3 resins does not improve obviously.

(5) Maybe, it is important to list the specific values (e.g., surface quality, shrinkage, heat

resistance, and toughness, etc.) for printed sample using proposed resin.

(6) The printed products shown in the paper are macro-sized samples, how about the printing effect of micro-sized structures. The authors should point out the minimum feature size for printing with the proposed resin, which may also be very important.

(7) The authors are also suggested to compare the proposed resin with other reported similar resins (especially the mechanical properties) to emphasize the advantage of the proposed resin.

(8) Logical confusions. Follows are few examples:

a) p. 6, line 10: "Investigating the Light Orientation Mechanism of PB-5CB Resins by SLA 3D printing". There are four paragraphs in this section, which are, in order, the effect of printing direction on properties, fracture mechanism analysis, molecular orientation on fracture inhibition, and polymer orientation. I think the right order should be the effect of printing direction on orientation, polymer orientation, effect of printing direction on properties, fracture mechanism analysis

b) p. 11, line 13: "Investigating the influence of printing resolution and LC content" In this section, the effect of the content of LCs and the printing resolution are mixed to be discuss. And in the last few paragraphs, the effect of the content of LCs on the orientation and the mixing properties of other resins are added, making this section long and confusing. For example, p. 11, line18: "The LC content...", this paragraph is about the different contents of 5CB LCs. However, p.12, line 10: "First, mechanical properties", there are only the properties with different printing resolutions rather than the results of the influence by contents of 5CB LCs.

Minor comments:

(1) Unnecessary contents. These could be placed in SI or removed according to their importance to the article.

a) Some contents in the introduction. I think the logical chain of this part should be 3d printing -SLA- photosensitive resin - filler method -LCs filling method-5CB filler. There are many unnecessary contents in this long chain, such as the contents of 3D printing, the application procedure of LCs (phase separation procedure), the inner structure of 5CB.

b) p. 5, line 17: The part "Dispersibility and rheological properties of PR-5CB resins". These contents fit better in SI. Leave a few sentences about these contents in the article is enough.

(2) Mistakes and omissions. Follows are few examples:

a) p. 4, line 17: "In the absence of photon absorption, the laser power required for the photoelectric field to sense the molecular orientation change is 102–103 W/cm²." I think the writer have it exactly backwards. It should be "In the absence of photon absorption, the laser power required for the photoelectric field to make the molecular orientation change is 102–103 W/cm²."

b) p. 5, line 1: "The strategy includes three design steps:" I only see two steps in this paragraph. The contents follows "Finally" is the performance research which is unrelated to the 3D printing strategy.

c) p. 5, line 18: "Good dispersion is important for obtaining high performance of 3D printed samples of PR-5CB resins." Which performance? These should not be omitted.

d) p. 6, Fig. 1: In these pictures, the numbers "0, 3, 5, 7" after the "PR-5CB" are first appeared and no explanation for these numbers until the p11 in 18: "Samples printed with different contents of 5CB LCs (3%, 5%, and 7%) were used to study the effect of 5CB content on the orientation state."

e) p. 7, line 21: "To further confirm our conjecture, the samples were subjected to additional bending tests, impact strength, and hardness tests." Which figure correspond to the result of test should be indicated when mentioned.

f) p. 10, line 7: "Fig. 2 shows the POM images of the 3D printed sample surface and cross-section at the 0° and 45° directions." Fig. 2 here should be Fig. 4.

The following are our point-by-point responses.

Responses to Reviewer 1:

Thank you for your valuable suggestions. I have incorporated the required editorial changes in the manuscript and reconsidered the conclusions. Please note that the changes in the revised text are highlighted in red font. All of our responses are given as follows:

1. Minor correction – SLA stands simply for “stereolithography” not “stereo lithography appearance” as written in both the abstract and introduction.

Response: Thank you for your valuable suggestion. This has been changed and highlighted in red font in the revised text.

2. The authors are encouraged to remove the exaggerated statement of “unlimited potential” in the last sentence of the abstract.

Response: Thank you very much for the valuable suggestions. This has been removed from the text.

3. The direct reference to “CLIP” at the end of the first paragraph is somewhat misleading since it is not used by the authors. Instead, the authors are encouraged to incorporate this reference into a previous sentence (line 48 perhaps) to include a statement on the high manufacturing speed of SLA, without directly writing CLIP so-as to not mislead the reader.

Response: We agree with your comment and have added the following sentences to the revised manuscript. SLA is the only light-curing 3D printing technology that can print large-size models. The laser beam, under the action of a deflecting mirror, can scan on a liquid surface—the trajectory of the scan and the presence or absence of light are controlled by a computer. The liquid cures at the point of luminous contact, manufacturing the parts quickly over a period of several to tens of hours.

4. I find the second paragraph of the introduction on line 53 of page 2 to be inaccurate and contradictory to statements made by the authors in the previous paragraph. As such, I strongly encourage that the authors remove a number of statements here that are an attack on objects produced by SLA (lines 53-61), and to refocus on the opportunities to improve material properties beyond what has been achieved so far, which the authors begin to do on line 62 “Several studies...”. In actuality, the breadth in mechanical properties of parts produced by

SLA is quite good (see for example: <https://bomar-chem.com/markets/3d-printing> , <https://www.carbon3d.com/materials>), and what this article showcases is that LC additives represent a unique avenue to further improving these properties.

Response: Thank you for your recommendation. By comparing the properties of resin materials with those provided on the website, we found that the resin materials on the website are divided into three main categories. One category is the rigid 3D printing inks and resins, with tensile strengths ranging from 22.1 to 90.0 MPa and elongation at break values ranging from 3 to 34%. The other category is the flexible 3D printing formulations, with tensile strengths ranging from 0.3 to 14.5 MPa and elongation at break values from 90 to 400%. The third category is tough 3D printing inks and resins, with tensile strengths ranging from 30 to 42 MPa and elongation at break values ranging from 30 to 120%. The optimal tensile strength and elongation at break values of the PR-5CB resin in our study reached 121.2 MPa and 25.5%, respectively, owing to the resin properties provided on the website. This shows that adding a specific percentage of LC can effectively enhance the properties of the resin.

5. The introduction provides alignment mechanisms starting at the top of page 4 (line 86), but it was very difficult to follow. It was unclear what the driving force for thermal alignment would be for both the 5CB and polymer chains – is there precedent for polymer chain alignment in standard SLA processes? The authors then go on to describe the photoelectric field effect for molecular orientation, which requires 10^2 - 10^3 W/cm² light intensities, which is possible for laser-based SLA, but not projection-based processes capable of providing the fast speeds mentioned by the authors in the introduction (see comment 3). However, the type of SLA 3D printer was not explicitly specified (it is a Form 2 from Formlabs, which is a laser-based system – but that should be stated) nor was the light intensity used in this system specified – and that should also be included along with the spot size of this printer. On line 120 the authors then comment on the printing direction, eluding to the importance of this in the liquid crystal orientation. Thus, it is very confusing after reading the introduction to know what effect or effects may be contributing to the reported LC alignment in this process. As such, the authors are encouraged to better clarify this text and provide a schematic to this point to assist readers in understanding the important mechanisms at play.

Response: Thank you for your comment. The following sentences have been added to the revised manuscript to clarify your concerns and make it easier for the reader to understand the mechanism at play.

Formslab Forms 2 printer with a 405 nm wavelength laser light source, and a laser power and spot size of 250 mW and 140 μm , respectively, was used; the calculated laser intensity can reach $10^5\text{W}/\text{cm}^2$, which is higher than the $10^2\text{--}10^3\text{ W}/\text{cm}^2$ required for the orientation of the 5CB liquid crystal we selected.

LCs can respond to the effect of temperature, light and electricity; therefore, this study proposes that during the SLA process, in the absence of photon absorption, the laser beam provides a photoelectric field that changes the molecular orientation in the printing engineering. To the authors' knowledge, there are no reports on the polymer orientation of products during the SLA process. Figure 1 shows the proposed mechanism of LC orientation during the SLA process. First, a proportion of LC is added to the photosensitive resin and dispersed by shear dispersion (Fig. 1(a)). Subsequently, curing of the photosensitive resin is performed, during which, the solubility of LC in the polymer matrix decreases, resulting in the agglomeration of LC particles—phase separation occurs. At this point, the curing time of each layer is controlled by adjusting the thickness of the print layer (Fig. 1(b)). Shortening of the curing time of each layer results in the fixation of LC particles to the print layers before aggregation can occur. Thus, finding the right print layer thickness can prevent the aggregation of LC and the generation of phase separation. Finally, the acrylate group of the photosensitive resin (Fig. 1(c)) will emit heat during the reaction process. Simultaneously, the laser power will produce a photoelectric field that will cause the LCs to undergo a phase transition to a specific orientation. This change will drive the polymer chains into the same orientation, resulting in their alignment

Fig. 1: Liquid crystal orientation during the SLA 3D printing process. (a) Preparation of LC-containing photosensitive resin. (b) Changing print layer thickness to control LC morphology. (c) LC orientation driving polymer orientation during printing.

6. At the beginning of the results and discussion section the authors are encouraged to clearly state what commercial resin was selected for this study and why that one in particular was selected. Additionally, the authors should state how much 5CB is added to the resin and why that amount was chosen at this stage (it only becomes apparent much later).

Response: We agree with your suggestion and have added the following text to address clarify this aspect of the study. Currently, most of the common photosensitive resins in the market are various types of acrylates, such as epoxy acrylates, urethane acrylates, and polyester acrylates. A general SLA photosensitive resin formulation was selected; that is, aliphatic polyurethane acrylate (CN9010), aliphatic polyester (CN991) based on a polyurethane diacrylate oligomer, ethylene oxide pentaerythritol tetraacrylate (SR494) as a prepolymer, and appropriate additives (HEMA) and initiators (TPO) were added. These materials were mixed, producing a low cost and high strength general-purpose photosensitive resin.

“This resin allows for the assessment of the effect of LC additives on the performance of photosensitive resins. Additionally, it provides the opportunity for subsequent mechanistic studies to improve the performance of photosensitive resins. Regarding the quantity of added 5CB, experiments have narrowed the required amount to a range of values; therefore, 3%, 5%,

and 7% of 5CB were added to LCs in this study. The addition of a small amount of 5CB can significantly improve the mechanical properties of the resin, ensuring a low cost of production and improving its practical applicability.”

7. The authors note the intensity of irradiation for the rheology to be “80 MW/cm²”, however this is likely a typo and should be “mW/cm²” – the authors should clarify this.

Response: You are correct. This typo was corrected in the revised text.

8. The authors claim that the presence of the 5CB in the resin increases the curing speed of the resin, but the difference shown does not appear statistically significant, nor do the authors provide standard deviations for this claim and thus it should be removed.

Response: We agree with the reviewer’s comment. This has been changed to state that the light-curing speed of the PR-5CB-3 resin was marginally higher than that of the commercial resin.

9. Similar to the above point, the authors are strongly encouraged to provide +/- 1 standard deviation from the mean of each measurement reported. According to the experimental section >3 samples were used for each mechanical measurement and as such this data should be available and will provide significance to the reported results. This is important as mechanical properties can be quite defect-dependent, leading to large variability.

Response: Thank you for your recommendation. We have revised Fig. 3 and Fig. 6 in the revised manuscript.

10. The claim that LC alignment along the stacking direction improves mechanical properties based on the results shown Figure 2B does not seem substantiated to me as the green and blue curves appear quite similar, and all four show somewhat unusual stress-strain responses for a rigid plastic, where there is an initial low Young’s modulus followed by a sharp increase in stiffness at different strains ~5-7%. The authors should comment on this unusual behavior. Furthermore, the difference in mechanical properties follow what is typical for printing horizontal vs vertical layers, where the larger difference observed for the 5CB cases could simply be ascribed to the increased heterogeneity, as observed in the optical data of Figure 3, leading to worsened layer-to-layer bonding. Thus, more evidence to support this claim surrounding the effects of molecular anisotropy on mechanical anisotropy should be provided.

Response: Thank you for your valuable suggestions. During the tensile test, a preload force is set to start the stretching process. This preload force is used by the sensor to begin its force calculations; thus, the stress-strain curve gradually and gently increases.

“To further verify the mechanism, the samples produced using different printing methods were subjected to surface and cross-sectional XRD tests (Fig. 4). The results were in agreement with those of POM as follows: first, the low surface and cross-sectional XRD peaks of the PR-5CB-0// and PR-5CB-0 \perp samples were the same. This indicated that the polymer aggregation morphology of PR-5CB-0 was not greatly affected by the change in printing direction. Additionally, the surface and cross-sectional polymer morphologies of the samples were the same, indicating that the polymer was not anisotropic. However, the surface XRD peaks of PR-5CB-0// and PR-5CB-3 \perp samples were significantly higher than those of the cross-sectional XRD peaks, suggesting that the crystallinity of the surface and cross-sectional aggregates were different, as were their aggregation states. The cross-sectional XRD peak of the PR-5CB-3 \perp sample was significantly higher than its surface XRD peak. Both showed higher crystallinity along the UV irradiation direction than that of the other directions, implying that the orientation of the polymer was along the UV irradiation direction.”

Fig. 4: XRD results for PR-5CB samples printed in vertical and horizontal directions.

11. The plots in figures 2 and 5 were very difficult to read. The authors are encouraged to increase the size of the plots and associated text to make them more legible.

Response: Thank you for your recommendation. We have made the necessary changes to Figs. 3 and 6 in the revised manuscript.

12. The paragraph starting on page 8 (line 207) describing the mechanism of fracture mechanics for brittle failure of highly crosslinked thermosets is unnecessary to include in the main text as it does not add insight for the specific system describe, but rather follows standard behavior for such materials. The authors are encouraged to remove this paragraph or put it in the supporting information file.

Response: We agree with the suggestion of the reviewer. The highlighted paragraph was deleted.

13. Paragraph 9 refers to the improved “toughness” of the 5CB containing samples, but “toughness” as it is being referred to here is not defined. The authors should define this and reference the associated data that corroborates the statements made about toughness enhancements in the presence of 5CB.

Response: Thank you for your recommendation. Impact strength is used to evaluate the impact resistance of a material or to determine the degree of brittleness and toughness of a material; therefore, the impact strength is also called impact toughness. Notch toughness is the ability of a material to absorb energy during the process of plastic deformation and fracture when it has a notch; this is a comprehensive performance of strength and plasticity. Therefore, under certain conditions, energy, strength, and plasticity can be indicative of toughness. The notched impact strength from Fig. 5(f) can be used to compare the toughness of materials. The results imply that the notched impact strengths of PC-5CB-0//, PC-5CB-0L, PC-5CB-3//, and PC-5CB-0L are 4.03, 2.41, 11.09, and 9.98 kJ/m², respectively, indicating that the addition of 5CB can effectively improve the toughness of 3D-printed products.

14. The SEM images in Figure 3 clearly show a difference between samples with and without 5CB, but it is not clear from these images why the striations are present and what direction the different samples are being viewed at. These points should be clarified.

Response: Thank you for your recommendation. Please see the following text that has been added to the revised manuscript to address the points raised by the reviewer:

“With the addition of 3% content 5CB, tough nesting and river-like cross-sections appeared in the cross-section of PR-5CB-3. Furthermore, a branching extension of microcracks,

which showed whitening, was evident, and the extension of microcracks ended with a typical river-like cross-section. Regarding the PR-5CB-3 \perp cross-section, the surface granularity was clearer, and its appearance was also river-like. This is mainly because 5CB produces a one-dimensional orientation in the printing process, and the PR-5CB-3// specimen is cracked by impact. This occurs because the impact direction is perpendicular to the direction of 5CB and polymer orientation, resulting in the formation of a “bridging zone” in the fractured area to prevent crack expansion and improve the impact strength of the material. Conversely, when the PR-5CB-3 \perp specimen was impacted, the 5CB and polymer orientation direction was parallel to the fractured surface. Therefore, the “bridging zone” could not be formed, resulting in a reduced impact strength enhancement effect compared to that of the PR-5CB-3// specimen. This result shows that, without the addition of 5CB, the difference in print orientation has little effect on the morphology of the cross-sectional fracture. However, the addition of 5CB effectively enhances the tough fracture of the 3D-printed product, and the change in print orientation affects the surface morphology of the fracture. The above analysis was consistent with the results of the mechanical property tests.”

Fig. 6: SEM images of the PR-5CB products printed in vertical and horizontal directions.

15. Line 244 references “Fig. 2”, but it should be “Fig. 4”.

Response: Thank you for identifying that error. This has been corrected in the revised manuscript.

16. The authors claim that the polymer chains are aligned based on the results of Figure 4 because it goes from fully light to dark upon rotating 45 degrees between crossed polarizers, as

opposed to only seeing “bright lines” (line 256). However, the magnification of the images shown is not provided and proper controls are missing with respect to POM images at the same magnification within the resin at different temperatures. If 5CB is dispersed homogeneously during the printing process then it seems expected to observe uniform changes in brightness under crossed polarizers.

Response: Thank you for your comment. To verify that 5CB can induce a certain orientation of the polymer during light curing, POM images of the surface of the PR-5CB-3// resin at different temperatures are shown in Supplementary Fig. 2.

“The PR-5CB-3//(25 μm) sample was observed using POM while being warmed at 30, 60, 90, 120, and 150 $^{\circ}\text{C}$ (Supplementary Fig. 2). By observing the POM images of PR-5CB at 10 \times 40 magnification, a clear change from completely dark to completely bright during the rotation of 0 $^{\circ}$ –45 $^{\circ}$ between cross-polarizers was observed. This change in characteristics proves that the polymer chain segments are anisotropic, indicating that some orientation of the polymer chains occurs. This is because thermotropic LCs (5CB), which have a unique protruding rod-like structure, reversibly change the spatial orientation of their bodies as the local temperature of the molecule is adjusted by the increase in temperature. LCs can transition from the randomly oriented state of the isotropic phase to the aligned state of the isotropic phase when below the isotropic transition temperature (30 $^{\circ}\text{C}$). Additionally, their POM maps appear to alternate between light and dark in a 360 $^{\circ}$ rotation between cross-polarizers. When the temperature of the LCs is raised above the isotropic transition temperature, 5CBs lose orientation order, resulting in POM diagrams that are always dark. However, the POM images of the polymer is always observed to be distinctly light and dark even when 5CB loses its oriented order as the temperature rises. This distinctive feature of the change is made further evident by the extent of the orientation of polymer chain segments.”

Supplementary Fig. 2: POM images of the PR-5CB-3// resin surface at different temperatures

17. Line 280 – the authors state that “the length of 5CB LCs was found to be roughly between 13 and 20 μm based on the POM images discussed above”, but do not reference which images they are talking about nor how those values were determined.

Response: Thank you for your comment. The length of 5CB LCs was found to be roughly between 13 and 20 μm based on the POM images described in Fig. 1. The POM diagram in Fig. 1 of the 5CB length was automatically measured and given a range when the instrument is tested. We can also calculate the actual length of the liquid crystal based on the length and magnification of the liquid crystal in the POM diagram (modified Fig. 1).

18. Line 302 – the authors refer to an increase in both strength and toughness, but again do not define toughness. If they are referring to the strain energy density (area under stress-strain curve) then this is not a true measure of toughness, which refers to the amount of energy required to propagate a crack, and thus requires an initial defect to be present during measurement.

Response: Thank you for your comment. Please see our response to the reviewer’s comment #13 as we believe it clarifies your concerns regarding the measure of toughness used in the study.

19. Figure 5 provides error bars for part (d), but they do not appear on the other bar graphs. The authors are encouraged to clarify what these error bars represent (e.g., are they ± 1 standard deviation?) and to include them on the other bar graphs to provide statistical significance.

Response: We agree with the reviewer’s comment. Figure 5 has been edited to clarify this issue in the revised manuscript. Mean values of multiple measurements are used to plot the

graphs of Figs. 5 (d), (e) and (f). We analyzed the data according to the origin software's averaging of multiple points data under the same conditions; additionally, the standard deviations were calculated and added as error bars to the plots.

20. The authors claim that polymer chains are oriented based on the fact that dichroism is observed after washing the sample with dichloromethane. However, the authors do not provide evidence that this washing actually removes the 5CB. If it does, then this would raise concerns about leaching of 5CB over time and a change in properties. This is also contradictory to the statement made by the authors in the introduction (lines 71-72) that the “LCs within the polymer are tightly and stably enclosed.”

Response: Thank you for your recommendation. First, to demonstrate that the polymer also undergoes orientation, we removed the small liquid crystal 5CB from the PR-5CB-3//(25 μ m) printed product with dichloromethane (DCM). After many hours of immersion, the 5CB was removed by DCM extraction until the mass of the sample remained constant. The extraction process reduced the weight of the product from 1.3984 g before extraction—to 1.3556 g after extraction—representing a weight loss of approximately 3%. This was consistent with our addition of 3%. This indicates that the method can effectively remove the 5CB liquid crystal from the 3D-printed products. The PR-5CB-0//(25 μ m)-printed products were extracted by dichloromethane for the same length of time; however, the weight did not change. This shows that DCM can remove 5CB without changing the 3D-printed products.

The reviewer raised the concern that 5CB would leach out over time and change the properties of the 3D-printed products. Here, we explain that 5CB acts as an inducer in the light curing process, triggering photosensitive polymerization along the direction of UV irradiation, thus anchoring 5CB within the polymer; therefore, it is difficult for 5CB to leach out of the polymer without specific solvent treatment. Alternatively, the highlight of our study is that 5CB is induced to align the polymer along a certain direction, making the polymer anisotropic. The performance of 3D-printed products is mainly determined by the nature of the polymer. The above extraction experiments show that solvent treatment has no effect on the polymer; therefore, if a small quantity of 5CB is leached out, the performance of the polymer should not be affected.

21. Figure 6 provides a schematic hypothesis as to changes in alignment for prints made with different thicknesses, but the evidence for this is solely dichroism observed under POM. Additional evidence, such as x-ray scattering would help to substantiate these claims. Also, the authors should detail what printing direction is being represented in Figure 6.

Response: Thank you for your valuable suggestions.

“X-ray diffraction can accurately describe the arrangement of LC molecules. For nematic LCs, no diffraction peaks appeared in the small-angle region; however, a dispersion peak appeared in the wide-angle region of $2\theta = 20^\circ$. The size, height, and width of this peak were related to the content of this phase in the sample, degree of dispersion, and orderliness. Generally, the higher and larger the peak, the more (orderly) surface phases are present. Therefore, the XRD of the thin surface section of the 3D-printed PR-5CB-0// and PR-5CB-3// samples at 25, 50, and 100 μm printing resolutions was tested to verify the orderly variation of LC (Fig. 10). The results show that there was little variation in the peaks of the RR-5CB-0-printed products at different printing resolutions. However, the XRD peak of the PR-5CB-3 sample at a printing resolution of 25 μm was higher than those of the PR-5CB-0 sample. The thickness of the PR-5CB-3 sample was approximately the same as the length of the rod LC and the molding time was short. Therefore, the LC oriented the surrounding photosensitive resin in the same direction as that of light irradiation. When the photosensitive resin polymerized, it fixed the 5CB in one direction, resulting in the polymer exhibiting anisotropy. Increasing the printing resolution to 50 μm resulted in a marginal decrease in the XRD peak. Because the

thickness of the polymer was greater than the length of the LC rod, the LC also aligned with the surrounding photosensitive resin during light orientation. However, the orientation was incomplete when the quantity of photosensitive resin was too high. When the printing resolution was increased to 100 μm , the height of the XRD peak significantly decreased, and the 5CB LC was surrounded by the photosensitive resin. Consequently, the photosensitive resin was dispersed as it could not be oriented during light treatment. This resulted in a relatively random orientation that the polymer could not adopt.

The effect of different contents of 5CB on the polymer orientation during 3D printing was examined. The XRD curves of PR-5CB-3//(25 μm), PR-5CB-5//(25 μm), and PR-5CB-7//(25 μm) were compared—the height of the XRD peaks decreased as the 5CB content increased. In particular, when the 5CB content reached 7%, the height of the XRD peak sharply decreased. This implies that an increase in 5CB content results in LC aggregation, making it more difficult for the LCs and the photosensitive resin to orient with light irradiation.”

Fig. 10: X-ray diffraction of PR-5CB resins at different print resolutions and contents.

22. The mechanism described on page 16 with respect to Figure 6b is very detailed, but no evidence for the formation of “spherical liquid crystals...fixed on the surface” is provided to support this text.

Response: Thank you for your comment. Two methods were used to verify that the spherical liquid crystals were immobilized on the surface of the samples and transformed into microcrystalline structures after cooling.

“The first method selected the surface film of PR-5CB-7 (50 μm) and soaked the film in dichloromethane for 6 h to remove the 5CB liquid crystal; subsequently, it was subjected to POM observation (Supplementary Fig. 3). The surface film of PR-5CB-7 (50 μm) with 5CB present is shown in Supplementary Figs. 3(a) and (b); this film has a quantity of aggregated spherical LC on its surface. However, the aggregated spherical LCs were not observed on the surface film of PR-5CB-7 (50 μm) with 5CB absent (Supplementary Figs. 3(c) and (d)). Because of an increase in LC content, the “anchor point” provided by the photosensitive resin was not able to induce partial LC orientation, resulting in the retention of 5CB in the form of droplets. Regarding the 3D-printed PR-5CB-7 sample, the lower surface orientation and LC aggregation resulted in a significant decrease in tensile strength and modulus. The droplet form of 5CB was not conducive to the improvement of product strength.

In the second method, the surface film of PR-5CB-7 (50 μm) was selected and placed on a heating table for concomitant heating and POM imaging—the results are shown in Supplementary Fig. 3(e). As the temperature increased, the spherical crystal aggregates on the surface of the film gradually disappear until they become completely black. This indicates that the spherical crystals slowly change from the crystalline state to the disordered state. This further verifies the aggregation of 5CB during the printing process when it is added in surplus to the resin.”

Supplementary Fig. 3: (a) POM images of PR-5CB-7//($50\ \mu\text{m}$), with and without the extraction of 5CB LC. (b) POM images of the surface of the PR-5CB-7//($50\ \mu\text{m}$) resin at different temperatures.

23. With respect to the resin selection made above (point 6), the authors claim that the effect of adding 5CB to resins is general and then reference results in the supporting information, specifically noting in the text that optical anisotropy is observed for three additional resin systems (line 414 in the main manuscript and Fig. S2 in the SI). However, the authors fail to comment on the fact that mechanical properties for all three other examples appear unaffected by the incorporate of the 5CB (Fig. S1), which is in contrast to the one shown in the main manuscript. This begs the question as to why this resin is particularly special and needs to be discussed by the authors as it goes against the generality of this approach.

Response: The reviewer has raised an interesting concern. However, the results of the formulations chosen for the paper and the three formulations in the SI support our conclusions. To ensure that our results are clear to our readers and reviewers, we stated the tensile results of each formulation in Supplementary Table 2. This table shows that adding 3% of 5CB can effectively improve the tensile properties of 3D-printed products. Additionally, the results of

the three formulations are consistent with the results of the paper. First, the addition of 3% 5CB can improve the tensile properties of 3D-printed products. The tensile strength of formulation F2 increased from 8.5 MPa in F2-5CB-0// to 15.0 MPa in F2-5CB-3//; additionally, the elongation at break also increased from 198.8% to 351.6%. Regarding formulation F3, the tensile strength increased from 12.5 MPa in F3-5CB-0// to 22.2 MPa in F3-5CB-3//, and the elongation at break increased from 146.2% to 222.0%. For formulation F4, the tensile strength increased from 11.8 MPa in F4-5CB-0// to 21.0 MPa in F2-5CB-3//, and the elongation at break increased from 52.2% to 84.8%. Second, the performance of 3D-printed products differed when printed in different directions. As discussed in our manuscript, the performance of vertical printing was better than that of horizontal printing. The tensile performance of formulation F2-5CB-3 \perp was 11.9 MPa and 313.6%, while that of F2-5CB-3// was 15.0 MPa and 351.6%. The tensile performance of formulation F3-5CB-3 \perp was 20.7 MPa and 200.1%, while that of F3-5CB-3// was 22.2 MPa and 222.0%. The tensile properties of formulation F4-5CB-3 \perp were 15.4 MPa and 71.3%, while those of F4-5CB-3// were 21.0 MPa and 84.8%.

Supplementary Fig. 6: Stress-strain curves of different commercial printing formulations with 3% content of 5CB.

Supplementary Table 2 Tensile strength and elongation at break values of the 3D printing strips based on F2, F3, and F4.

		Tensile Strength/MPa	Elongation at break/%
F2	F2-5CB-0//	8.5	198.8
	F2-5CB-3 \perp	11.9	313.6
	F2-5CB-3//	15.0	351.6

F3	F3-5CB-0//	12.5	146.2
	F3-5CB-3⊥	20.7	200.1
	F3-5CB-3//	22.2	222.0
F4	F4-5CB-0//	11.8	52.2
	F4-5CB-3⊥	15.4	71.3
	F4-5CB-3//	21.0	84.8

“To further demonstrate that the strategy of light-oriented 3D printing of 5CB LC/photocurable resins is effective across all acrylic photocurable resins, the following tests were performed. First, different commercial printing formulations (Supplementary Table 1) with 3% 5CB content were selected; sample strips of these formulations were printed at 25 μm print resolution to test their tensile properties (Supplementary Fig. 6 and Supplementary Table 2) and examine their POM images (Supplementary Fig. 7). Alternating four bright and four dark images were observed for all printing formulations when rotated by 360° using the carrier table (Supplementary Movies 8–10). These results demonstrate that all photosensitive resins with added 5CB are well-oriented when exposed to light, which improves their mechanical properties.

Second, experiments were designed to investigate the role of various prepolymers in the light orientation of 5CB samples by designing multiple formulations, preparing various types of photosensitive resins containing 3% content of 5CB (see Supplementary Table 3 for the formulations), and conducting POM tests on the printed samples—the results are shown in Supplementary Fig. 8. Alternating four bright and four dark images were also observed for all formulations, indicating that they were all oriented by the behavior of 5CB. This verifies that if the modified resin has acrylate photosensitive groups, it can be oriented by the behavior of 5CB.”

Supplementary Table 3: Formulation of various prepolymer/3% 5CB photosensitive resins.

F4	CN991	HPMA	TPO	
	62	35	3	
F5	Aliphatic polyether acrylates	HEMA	TPO	
	67	30	3	
F6	CN9010	HEMA	TPO	
	58	39	3	
F7	Aliphatic polyether acrylates	HEMA	HPMA	TPO
	50	20	27	3
F8	Aliphatic polyether acrylates	CN991	HEMA	TPO
	30	35	32	3
F9	CN9010	HEMA	HPMA	TPO
	57	10	30	3
F10	CN9010	CN991	HEMA	TPO
	32	28	37	3
F11	SR494	HEMA	HPMA	TPO
	50	30	17	3

Supplementary Fig. 8: Polarizing optical microscopy (POM) of different prepolymer printing formulations with a 3% content of 5CB. Images were taken at angles of 0° and 45°.

24. Figure 10a shows differences in haze across a 3D object that was uniformly printed either parallel or perpendicular. The authors are encouraged to comment on where this observed contrast arises from when looking at what appears to be planar (flat) objects as in part (a).

Response: Thank you for your recommendation.

“The LCs were oriented along the printing direction when the printing directions were vertical (PR-5CB-3//) and horizontal (PR-5CB-3⊥) (Fig. 7). However, planar (flat) observations showed that the orientation of the 5CB LC and polymer chains in the PR-5CB-3// samples were perpendicular to visible light; additionally, upon visible light irradiation, the 5CB and polymer chains would reflect the light, limiting penetration of the printed model by visible light, resulting in a milky coloration. When the PR-5CB-3⊥ model, the orientation of the 5CB LC and polymer chains were parallel to visible light. Upon visible light irradiation, the 5CB and polymer chain reflected less light than that of the PR-5CB-3// samples, which allowed the passage of visible light along the orientation of its 5CB LC and polymer chains, resulting in its transparency.

Fig. 7: Visual difference between the vertical and horizontal directions of the printed model.

25. Line 489 – the authors claim that the printing accuracy of the teeth was “0.14-0.28%”, but it is not clear where this value comes from and how it was measured. The authors are encouraged to expand upon this statement.

Response: Thank you for your comment. Please see the following text, which has been included in the manuscript, for clarification of these values.

“Figure 14(a) shows the gear model of the PR-5CB-3// resin at different printing resolutions. The diameter of the gear stereolithography (stl.) file is 70.00 mm and the formulated PR-5CB-3// resin with the best mechanical properties for printing was chosen. The diameter of the PR-5CB-3//(25 μ m) resin printout was 69.98 mm with a relative error of 0.03%; that of the PR-5CB-3//(50 μ m) resin was 70.08 mm with a relative error of 0.12%; and that of the PR-5CB-3//(100 μ m) resin was 70.21 mm with a relative error of 0.30%. To verify that the printed models did not deform after high temperature treatment, the PR-5CB-3//(25 μ m) printed samples were heat treated at different temperatures for 24 h. The printed products did not deform as the temperature increased, and sample shrinkage at high temperatures remained below 0.05% (Fig. 14(b)). Figures 14(c) and (d) show that the models with high precision requirements were printed using the PR-5CB-3 resin. The details of each model were clearly visible, illustrating the potential use of this photosensitive resin in producing advanced precision instruments. The skeletonized sphere remained structurally intact as observed at different magnifications, indicating that high quality can be achieved. Micro-sized structures are also obtained, and the minimum skeletonized sphere size that can be printed is up to 3 mm in diameter. Overall, these results demonstrate that the PR-5CB resin prepared in this study has high printing resolution and can be used for printing various products with high accuracy requirements.”

Fig. 14: (a) Shrinkage of PR-5CB-3// resin at different resolutions. (b) Printing model of PR-5CB-3//($25\ \mu\text{m}$) treated at different temperatures. (c) Skeletonized spheres with a resolution of 2833×2918 pixels, and height of $150\ \mu\text{m}$ magnified by $10\times$ and $50\times$. (d) Printing model of macro-sized and micro-sized structures.

Responses to Reviewer 2:

Thank you for your valuable suggestions. I have incorporated the required editorial changes in the manuscript and reconsidered the conclusions. Please note that the changes in the revised text are highlighted in red font. All of our responses are given as follows:

1. The language needs to be polished; the writing is colloquial and the words are not rigorous and academic; some pictures are not clear; scale bar is missing, too many typos, and the caption is not specific enough.

Response: Thank you for your comment. The manuscript has been professionally edited for language and grammar, and the figures, as well as their captions, have been improved for clarity.

2. The arrangement of LC in the printed layer in Figures 4 and 6 should be introduced in Figure 2, which is crucial to help readers understand the difference in mechanical properties. In addition, why are LC molecules aligned vertically to the upper and lower surfaces of the gap (Fig. 6b)? Normally, the long axes of LC molecules tend to be aligned parallel to the upper and lower surfaces of the gap rather than vertically. More scientific explanation is needed.

Response: We appreciate the reviewer's comment.

“A closed environment restricts the mobility of LCs; therefore, the LCs must interact with the surrounding interface. This will affect the arrangement state of the LC molecules in the absence of external conditions. Generally, if the force between the LC molecules is greater than the interaction between the LCs and the interface, the LC molecules tend to align perpendicular to the surface; conversely, if the surface energy of the substrate is greater than the surface energy of the LC molecules, they tend to align parallel to the surface. Fourier transform polarization infrared spectroscopy (FTIR) can be used to confirm the alignment pattern of the LC molecules. The -CN group in the 5CB molecule, which has a strong vibrational absorption at approximately 2226 cm^{-1} , was chosen as the “probe.” The vibrational direction of -CN coincided with the long-axis direction of the LC. Therefore, measuring the polarized infrared absorption spectrum of -CN can detect the alignment direction of LC. Supplementary Fig. S4 (a) shows the printed test films, based on varying printing methods, with film thicknesses of approximately $150\text{ }\mu\text{m}$. The printed films were subjected to FTIR tests, where P is the polarization direction of the IR polarized light and E is the vibration vector of the polarized UV light. The vibrational absorption of -CN was strong when P//E, but weak when P⊥E (Supplementary Fig. 4(b)). This suggests that the LC molecules preferred to align along the direction parallel to the vector of UV light vibrations.”

Supplementary Fig. 4: (a) Different methods used for printing IR test films; (b) FTIR spectra of the films.

3. Line135: "a large number of rod-shaped LC molecules", is it a single molecule or grouped molecules? Please clarify.

Response: Thank you for your comment. The 5CB liquid crystal molecules are shaped like cylinders with long axes of molecules that are almost parallel to each other. Each rod-like structure in Fig. 2(a) is an aggregate of multiple 5CB liquid crystal molecules.

4. Line342: "extracting 5CB LC", the authors didn't mention any extraction procedure in the Methods section. Please add it if it is necessary.

Response: Thank you for your recommendation. We have added the required text in the revised manuscript. Dichloromethane (DCM) was used to remove the small LC (5CB) from the PR-5CB printed product. After many hours of immersion, the 5CB was removed by DCM extraction until the mass of the sample remained constant.

5. Line393: If the red arrow rotates, then the picture should not be rotated. Why is the POM picture of PR-5CB-0 bright at when the thickness is 50 microns?

Response: Thank you for your comment. We repeated the test and modified Fig. 9 using the updated results.

6. Line470: "When the printing direction is parallel to the layer stacking direction, the liquid crystal orientation direction is perpendicular to the printing direction, and the model shows a higher haze with a milky white color." Isn't the printing direction always perpendicular to the printing layer? Please clarify.

Response: Thank you for your question. The visual differences between the printed models were observed, as shown in Fig. 7.

“The LCs were oriented along the printing direction when the printing directions were vertical (PR-5CB-3//) and horizontal (PR-5CB-3⊥) (Fig. 7). However, planar (flat) observations showed that the orientation of the 5CB LC and polymer chains in the PR-5CB-3// samples were perpendicular to visible light; additionally, upon visible light irradiation, the 5CB and polymer chains would reflect the light, limiting penetration of the printed model by visible light, resulting in a milky coloration. When the PR-5CB-3⊥ model, the orientation of the 5CB LC and polymer chains were parallel to visible light. Upon visible light irradiation, the 5CB and polymer chain reflected less light than that of the PR-5CB-3// samples, which allowed the passage of visible light along the orientation of its 5CB LC and polymer chains, resulting in its transparency.”

Fig. 7: Visual difference between the vertical and horizontal directions of the printed model.

7. Line519: "Aliphatic urethane acrylate (CN9010), polyurethane acrylate (CN991) ethoxylated pentaerythritol tetraacrylate (SR494)", why are four polymers needed for the fabrication? Please clarify their functions in the light orientation of LCs?

Response: Thank you for your question.

“Currently, most of the common photosensitive resins in the market are various types of acrylates, such as epoxy acrylates, urethane acrylates and polyester acrylates. A general SLA photosensitive resin formulation was selected; that is, aliphatic polyurethane acrylate (CN9010), aliphatic polyester (CN991) based on a polyurethane diacrylate oligomer, ethylene oxide pentaerythritol tetraacrylate (SR494) as a prepolymer, and appropriate additives (HEMA) and initiators (TPO) were added. These materials were mixed, producing a low cost and high strength general-purpose photosensitive resin.”

“To further demonstrate that the strategy of light-oriented 3D printing of 5CB LC/photocurable resins is effective across all acrylic photocurable resins, the following tests were performed. First, different commercial printing formulations (Supplementary Table 1) with 3% 5CB content were selected; sample strips of these formulations were printed at 25 μm

print resolution to test their tensile properties (Supplementary Fig. 6 and Supplementary Table 2) and examine their POM images (Supplementary Fig. 7). Alternating four bright and four dark images were observed for all printing formulations when rotated by 360° using the carrier table (Supplementary Movies 8–10). These results demonstrate that all photosensitive resins with added 5CB are well-oriented when exposed to light, which improves their mechanical properties.

Second, experiments were designed to investigate the role of various prepolymers in the light orientation of 5CB samples by designing multiple formulations, preparing various types of photosensitive resins containing 3% content of 5CB (see Supplementary Table 3 for the formulations), and conducting POM tests on the printed samples—the results are shown in Supplementary Fig. 8. Alternating four bright and four dark images were also observed for all formulations, indicating that they were all oriented by the behavior of 5CB. This verifies that if the modified resin has acrylate photosensitive groups, it can be oriented by the behavior of 5CB.”

Supplementary Table 3 Formulation of various prepolymer/3% 5CB photosensitive resins.

F4	CN991	HPMA	TPO	
	62	35	3	
F5	Aliphatic polyether acrylates	HEMA	TPO	
	67	30	3	
F6	CN9010	HEMA	TPO	
	58	39	3	
F7	Aliphatic polyether acrylates	HEMA	HPMA	TPO
	50	20	27	3
F8	Aliphatic polyether acrylates	CN991	HEMA	TPO
	30	35	32	3
F9	CN9010	HEMA	HPMA	TPO
	57	10	30	3
F10	CN9010	CN991	HEMA	TPO
	32	28	37	3

F11	SR494	HEMA	HPMA	TPO
	50	30	17	3

Supplementary Fig. 8: Polarizing optical microscopy (POM) of different prepolymer printing formulations with a 3% content of 5CB. Images were taken at angles of 0° and 45° .

8. LC is a very good optical material. It is not worth to study the improvement of mechanical properties by aligning LC since it is quite expensive compared with other fillers which can reach higher mechanical enhancement. More innovative application related to programmable control of LC arrangement should be added. The redundant mechanical tests should be placed in the support information.

Response: We appreciate the reviewer's comment. However, some of the main reasons for selecting LC for this study were its universality and relatively cheap price. The market price of 5CB liquid crystal was approximately \$280/kg. The price range of printing resin for SLA forms2 (clear, white, Rigid 4000, Tough2000, rigid 10K resin) varied between \$126 to 364/kg. The addition of 3% of 5CB is equivalent to an increase in price of approximately \$8/kg. Therefore, this increases the cost of the print by approximately 2.1%–6.3%. This price increase is acceptable based on the performance improvement of the resin. Another reason for the selection of LC was that the principle behind the improved performance of the 3D-printed

products differed from that of other common additives, which usually act as an inorganic filler or as a prepolymer to modify 3D-printed products. However, the addition of 5CB improved the performance of 3D-printed products by changing the aggregation state of the polymer from random to somewhat ordered. This was an interesting finding in 3D printing. Of course, the implications of this finding must be further explored; in particular, the more innovative applications related to LC arrangement programmable control. We believe that there will be new breakthroughs in the near future. To emphasize the superiority of the proposed resin, we summarized the mechanical properties of other reported similar resins for comparison with the LC 5CB resins of this study in Supplementary Table 4.

9. Sentences are difficult to understand or have inappropriate terminology, some of them are listed below:

Response: We do apologize for any inappropriate terminologies or poorly constructed statements. Please see the changes that we have made to each response. We hope that any misconceptions have been clarified. Additionally, these changes have been made to the text where stated and highlighted in red font.

(1) Line21: "the light-driven orientation of 5CB drives other acrylate prepolymers to orient along the orientation direction of 5CB", hard to understand.

Response: This has been edited for clarity as follows: The light-driven orientation process of 5CB guides other acrylate prepolymers to orient in the same direction.

(2) Line24: What do you mean by "3D printing lamination method"?

Response: It is changed to 3D printing method. The 3D printing method refers to the vertical and horizontal printing methods.

(3) (4) Line29: Too many similar repeated expressions, like "121.2 MPa, 25.5%, 222.0 MPa, 11.09 kJ/m², and 1702.3 Mpa respectively; these values are 281%, 241%, 270%, 275%, and 186% of those of the commercial inks". They need a different expression.

Response: This has been edited for clarity as follows: The tensile strength, elongation at break, flexural strength, impact strength and storage strength of the 3D-printed products of PC-5CB-3/(25 μm) were 121.2 MPa, 25.5%, 222.0 MPa, 11.09 kJ/m² and 1702.3 MPa, respectively, which were 2.81-, 2.41-, 2.70-, 2.75- and 1.86-fold higher than the commercial resins.

(5) Line38: "molding method and diversified molding", the "molding" is not suitable to describe 3D printing.

Response: Thank you for your comment. Molding uniformly changed to printing.

(6) Line 57: "the formed parts prepared by the development of the photosensitive resin", hard to understand.

Response: Thank you for your comment. This has been rectified in the text.

(7) Line175: "PC-5CB-0 \perp ", typo of repeating

Response: This has been revised in the text.

(8) Line222: "Fig. 2.", typo

Response: All figure captions have been formatted based on the journal's guidelines. We have also ensured that any typos have been addressed.

(9) Line237: What is the specific position of the picture in the sample in Figure 3? Need a schematic diagram.

Response: Thank you for your comment.

“With the addition of 3% content 5CB, tough nesting and river-like cross-sections appeared in the cross-section of PR-5CB-3. Furthermore, a branching extension of microcracks, which showed whitening, was evident, and the extension of microcracks ended with a typical river-like cross-section. Regarding the PR-5CB-3 \perp cross-section, the surface granularity was clearer, and its appearance was also river-like. This is mainly because 5CB produces a one-dimensional orientation in the printing process, and the PR-5CB-3// specimen is cracked by impact. This occurs because the impact direction is perpendicular to the direction of 5CB and polymer orientation, resulting in the formation of a “bridging zone” in the fractured area to prevent crack expansion and improve the impact strength of the material. Conversely, when the PR-5CB-3 \perp specimen was impacted, the 5CB and polymer orientation direction was parallel to the fractured surface. Therefore, the “bridging zone” could not be formed, resulting in a reduced impact strength enhancement effect compared to that of the PR-5CB-3// specimen. This result shows that, without the addition of 5CB, the difference in print orientation has little effect on the morphology of the cross-sectional fracture. However, the addition of 5CB effectively enhances the tough fracture of the 3D-printed product, and the change in print

orientation affects the surface morphology of the fracture. The above analysis was consistent with the results of the mechanical property tests.”

Fig. 6: SEM images of the PR-5CB products printed in vertical and horizontal directions.

(10) Line274/337: Is it suitable to use "lamination","printing accuracy" in this article?

Response: Thank you for your comment. The use of "lamination" and "printing accuracy" in this article may not be quite correct, so change it to printing direction and printing resolution.

Responses to Reviewer 3:

Thank you for your valuable suggestions. I have incorporated the required editorial changes in the manuscript and reconsidered the conclusions. Please note that the changes in the revised text are highlighted in red font. All of our responses are given as follows:

1. Why is 4'-Pentyl-4-cyanobiphenyl (5CB) selected instead of other commonly LCs?

Response: Thank you for your question. There are two main reasons for choosing 4'-pentyl-4-cyanobiphenyl (5CB) as the liquid crystal for our study: first, it is one of the most studied LC materials because it is easy to obtain nematicity at room temperature. Additionally, SLA 3D printing is performed at room temperature; therefore, using this technique ensures that 5CB can be relatively well transformed in the nematic and isotropic states during the printing process. Second, 5CB is an inexpensive liquid crystal, making it affordable for a wide range of applications.

2. The orientation of LCs is the core for the proposed resin. The advantage should be emphasized both in “introduction” and “application display” again. At present, the printed sample, in manuscript, is too simple.

Response: Thank you for your suggestion. The following sentences have been added to the revised manuscript to clarify your concerns and make it easier for the reader to understand the advantage.

"LCs can respond to the effect of temperature, light and electricity; therefore, this study proposes that during the SLA process, in the absence of photon absorption, the laser beam provides a photoelectric field that changes the molecular orientation in the printing engineering. To the authors' knowledge, there are no reports on the polymer orientation of products during the SLA process. Figure 1 shows the proposed mechanism of LC orientation during the SLA process. First, a proportion of LC is added to the photosensitive resin and dispersed by shear dispersion (Fig. 1(a)). Subsequently, curing of the photosensitive resin is performed, during which, the solubility of LC in the polymer matrix decreases, resulting in the agglomeration of LC particles—phase separation occurs. At this point, the curing time of each layer is controlled

by adjusting the thickness of the print layer (Fig. 1(b)). Shortening of the curing time of each layer results in the fixation of LC particles to the print layers before aggregation can occur. Thus, finding the right print layer thickness can prevent the aggregation of LC and the generation of phase separation. Finally, the acrylate group of the photosensitive resin (Fig. 1(c)) will emit heat during the reaction process. Simultaneously, the laser power will produce a photoelectric field that will cause the LCs to undergo a phase transition to a specific orientation. This change will drive the polymer chains into the same orientation, resulting in their alignment in one direction. "

Fig. 1: Liquid crystal orientation during the SLA 3D printing process. (a) Preparation of LC-containing photosensitive resin. (b) Changing print layer thickness to control LC morphology. (c) LC orientation driving polymer orientation during printing.

Regarding the advantages in application display, we mainly embody two aspects. The first is that by changing the printing direction, printed samples with different transparency can be obtained. "The LCs were oriented along the printing direction when the printing directions were vertical (PR-5CB-3//) and horizontal (PR-5CB-3⊥) (Fig. 7). However, planar (flat) observations showed that the orientation of the 5CB LC and polymer chains in the PR-5CB-3//

samples were perpendicular to visible light; additionally, upon visible light irradiation, the 5CB and polymer chains would reflect the light, limiting penetration of the printed model by visible light, resulting in a milky coloration. When the PR-5CB-3 \perp model, the orientation of the 5CB LC and polymer chains were parallel to visible light. Upon visible light irradiation, the 5CB and polymer chain reflected less light than that of the PR-5CB-3 \parallel samples, which allowed the passage of visible light along the orientation of its 5CB LC and polymer chains, resulting in its transparency.

Fig. 7: Visual difference between the vertical and horizontal directions of the printed model.

The second is further elaboration for printing sample surface quality, heat resistance, shrinkage rate and micro printing. Please see our response to the reviewer's comment #5 as we believe it clarifies your concerns

3. Will the orientation of LC molecules trigger the contraction of printed sample if temperature exceeds phase transition temperature like previous studies (such as ACS applied materials & interfaces 13(2020): 12698-12708; Advanced Materials 33(2021): 2002541; Science advances 6(2020): eabc0034.)? If the printed sample generates evidently deformation

at high temperature (i.e., exceed phase transition temperature), the application of it will be restricted.

Response: Thank you for your question. As stated by the reviewer, all three mentioned papers were innovative studies on 3D-printed liquid crystal elastomers (LCEs). The phase temperature intervals of the prepared liquid crystal elastomers were investigated based on the understanding that liquid crystal materials provide unique advantages to the development of the additive manufacturing process by controlling various property changes of liquid crystal polymers at different temperatures.

The liquid crystal content was set at 3%, 5% and 7% in this study; however, the effect of different temperatures on the deformation and performance of liquid crystal materials was almost negligible. To verify that the printed models did not deform after high temperature treatment, the PR-5CB-3//($25\ \mu\text{m}$) printed samples were heat treated at different temperatures for 24 h. The printed products did not deform as the temperature increased, and sample shrinkage at high temperatures remained below 0.05% (Fig. 14(b)).

4 The manuscript exaggerates the light-curing ability of the PR-5CB resins. From figure 1c, the light-curing speed of the PR-5CB-3 resins does not improve obviously.

Response: We agree with the reviewer's comment. This has been rectified in the text.

5 Maybe, it is important to list the specific values (e.g., surface quality, shrinkage, heat resistance, and toughness, etc.) for printed sample using proposed resin.

Response: Thank you for your comment. This has been stated in the text as follows:

“Figure 14(a) shows the gear model of the PR-5CB-3// resin at different printing resolutions. The diameter of the gear stereolithography (stl.) file is 70.00 mm and the formulated PR-5CB-3// resin with the best mechanical properties for printing was chosen. The diameter of the PR-5CB-3//($25\ \mu\text{m}$) resin printout was 69.98 mm with a relative error of 0.03%; that of the PR-5CB-3//($50\ \mu\text{m}$) resin was 70.08 mm with a relative error of 0.12%; and that of the PR-5CB-3//($100\ \mu\text{m}$) resin was 70.21 mm with a relative error of 0.30%. To verify that the printed models did not deform after high temperature treatment, the PR-5CB-3//($25\ \mu\text{m}$) printed samples were heat treated at different temperatures for 24 h. The printed products did not deform as the temperature increased, and sample shrinkage at high temperatures remained below 0.05% (Fig. 14(b)). Figures 14(c) and (d) show that the models with high precision

requirements were printed using the PR-5CB-3 resin. The details of each model were clearly visible, illustrating the potential use of this photosensitive resin in producing advanced precision instruments. The skeletonized sphere remained structurally intact as observed at different magnifications, indicating that high quality can be achieved. Micro-sized structures are also obtained, and the minimum skeletonized sphere size that can be printed is up to 3 mm in diameter. Overall, these results demonstrate that the PR-5CB resin prepared in this study has high printing resolution and can be used for printing various products with high accuracy requirements.”

Fig. 14: (a) Shrinkage of PR-5CB-3 resin at different resolutions. (b) Printing model of PR-5CB-3 (25 μm) treated at different temperatures. (c) Skeletonized spheres with a resolution of 2833 × 2918 pixels, and height of 150 μm magnified by 10× and 50×. (d) Printing model of macro-sized and micro-sized structures.

6. The printed products shown in the paper are macro-sized samples, how about the printing effect of micro-sized structures. The authors should point out the minimum feature size for printing with the proposed resin, which may also be very important.

Response: Thank you for your comment. The printing effect on micro-sized structures has been included in the text. Please see our response to the reviewer’s comment #5 as we believe it clarifies your concerns regarding the minimum feature size for printing with the proposed resin.

7. The authors are also suggested to compare the proposed resin with other reported similar resins (especially the mechanical properties) to emphasize the advantage of the proposed resin.

Response: Thank you for your suggestion. The mechanical performance data of some 3D-printed samples using different photosensitive resins are compiled in Supplementary Table 4. The data show that the tensile strength of commercially available photosensitive resins is generally not high. If fillers are added to increase the tensile strength, the toughness of the product will decrease. In contrast, the PR-5CB resins developed in this study have excellent mechanical and thermal properties. In addition, the tensile strength and toughness can be simultaneously enhanced. Considering the overall performance of the final product, PR-5CB-3// is the best-performing resin.

Supplementary Table 4: Comparison of mechanical and thermal properties of 3D-printed samples composed of varying photosensitive resins.

Samples	Tensile Strength (MPa)	Elongation at Break (%)	Reference
SiO ₂ filling	53.8	2.7	[27]
Calcium sulfate whiskers filling	29.0	4.8	[28]
Graphene oxide filling	61.9	7.2	[29]
polysiloxane core-shell nanoparticles	53.6	12.84	[31]
Multi-walled carbon	57.0	9.9	[32]
Polyimide	24.9	5.4	[33]
ALCR-2	65.3	17.3	[24]
PR-5CB-3//	121.2	25.5%	This work

8 Logical confusions. Follows are few examples: a) p. 6, line 10: “Investigating the Light Orientation Mechanism of PB-5CB Resins by SLA 3D printing”. There are four paragraphs in this section, which are, in order, the effect of printing direction on properties, fracture mechanism analysis, molecular orientation on fracture inhibition, and polymer orientation. I think the right order should be the effect of printing direction on orientation, polymer orientation, effect of printing direction on properties, fracture mechanism analysis

Response: We appreciate the reviewer’s suggestion. This has been rectified as suggested.

b) p. 11, line 13: “Investigating the influence of printing resolution and LC content” In this section, the effect of the content of LCs and the printing resolution are mixed to be discuss. And in the last few paragraphs, the effect of the content of LCs on the orientation and the mixing properties of other resins are added, making this section long and confusing. For example, p. 11, line18: “The LC content...”, this paragraph is about the different contents of 5CB LCs. However, p.12, line 10: “First, mechanical properties”, there are only the properties with different printing resolutions rather than the results of the influence by contents of 5CB LCs.

Response: Thank you for your comment. We have edited this section to improve its structure and flow. We believe its clarity has been improved based on your recommendations.

Minor comments:

(1) Unnecessary contents. These could be placed in SI or removed according to their importance to the article.

a) Some contents in the introduction. I think the logical chain of this part should be 3d printing -SLA- photosensitive resin - filler method -LCs filling method-5CB filler. There are many unnecessary contents in this long chain, such as the contents of 3D printing, the application procedure of LCs (phase separation procedure), the inner structure of 5CB.

Response: Thank you for your comment. We have made changes to the introduction. We hope that it rectifies your concerns regarding its flow.

b) p. 5, line 17: The part “Dispersibility and rheological properties of PR-5CB resins”. These contents fit better in SI. Leave a few sentences about these contents in the article is enough.

Response: Thank you for your comment. Because the dispersion and rheological properties of the photosensitive resin directly affect the printing effect and should be measured, this part is recommended to be kept in the original manuscript. However, we make certain modifications to the figures and results, which can be found in the red marked part of the text.

(2) Mistakes and omissions. Follows are few examples:

a) p. 4, line 17: “In the absence of photon absorption, the laser power required for the photoelectric field to sense the molecular orientation change is 10^2 – 10^3 W/cm².” I think the writer have it exactly backwards. It should be “In the absence of photon absorption, the laser power required for the photoelectric field to make the molecular orientation change is 10^2 – 10^3 W/cm².”

Response: Thank you for your correction. This has been added to the text.

b) p. 5, line 1: “The strategy includes three design steps:” I only see two steps in this paragraph. The contents follows “Finally” is the performance research which is unrelated to the 3D printing strategy.

Response: Thank you for your comment. It has been modified to two steps in the text.

c) p. 5, line 18: “Good dispersion is important for obtaining high performance of 3D printed samples of PR-5CB resins.” Which performance? These should not be omitted.

Response: Thank you for your question. This has been changed to the following:

Good dispersion is important for obtaining high-performing 3D-printed samples using PR-5CB resins. Therefore, it is important to study the rheological properties of the composites as influenced by additive content, as well as the effect of adding plasticizers and compatibility agents, which can improve feedback processability. Additionally, if the additive is not well bonded to the resin matrix, it will lead to a degradation of mechanical properties in the cured product. I hope this answers your question.

d) p. 6, Fig. 1: In these pictures, the numbers “0, 3, 5, 7” after the “PR-5CB” are first appeared and no explanation for these numbers until the p11 in 18: “Samples printed with different contents of 5CB LCs (3%, 5%, and 7%) were used to study the effect of 5CB content on the orientation state.”

Response: We appreciate the reviewer’s comment. We have edited the figure in the manuscript to ensure that its description is clear to the reviewer and the readers.

e) p. 7, line 21: “To further confirm our conjecture, the samples were subjected to additional bending tests, impact strength, and hardness tests.” Which figure correspond to the result of test should be indicated when mentioned.

Response: We agree with the reviewer’s comment and have ensured that the figures are correctly associated with the text.

f) p. 10, line 7: “Fig. 2 shows the POM images of the 3D printed sample surface and cross-section at the 0° and 45° directions.” Fig. 2 here should be Fig. 4.

Response: Thank you for your comment. All figure labels have been edited to ensure they are correctly associated with the text and to align with the journal’s formatting guidelines.

REVIEWER COMMENTS

Reviewer #1 (Remarks to the Author):

The authors, Zhuo, Chen, Chen, and co-workers, are commended for their significant edits to the manuscript. Overall, the authors have addressed most of the reviewer comments to a sufficient extent and the clarity of the text has been considerably improved. For example, the XRD data was a nice addition to support the claims. However, a few issues remain that should be addressed prior to publication as noted below.

Comments/questions:

- 1) On lines 46-47 of the revised manuscript, the authors write that "...manufacturing the parts quickly over a period of several to tens of hours." However, this statement is vague as it does not provide part dimensions associated with the printing time. More effective would be providing a printing rate in terms of curing volume/time.
- 2) The text in the image of Figure 3B is difficult to read, in part due to the poor color contrast and in-part due to the low resolution and small font size (e.g., scale bars).
- 3) Lines 212-213 the authors state "However, the surface XRD peaks of PR-5CB-0// and PR-5CB-3⊥..." – I think the authors meant for this to be PR-5CB-3 for both parallel and perpendicular orientations.
- 4) Figure 4 key appears to have a similar issue. The final sample in the list should be PR-5CB-3 not PR-5CB-0 as it is currently written.
- 5) Line 262: "...5CB, tough nesting and..." should I think be "...5CB, toughness and..."
- 6) The authors are encouraged to specifically note that the increased crosslink density is due to an increase in physical crosslinks. As the text stands it was not immediately obvious why crosslink density would increase by adding 3% 5CB to the resin as typically in vat photopolymerization crosslinks refer to covalent bonds as opposed to intermolecular interactions.
- 7) The authors have not sufficiently addressed my concerns with the stress-strain curves in Figure 5a, which show an initially low modulus, followed by a sharp increase. The pre-load force mentioned by the authors in the rebuttal should not account for such a large percentage as that observed for the soft regions in these measurements. Also, preloading is not typically included in the stress-strain plots. The other concern here is that there appears

to be no significant difference in tensile modulus for samples oriented parallel vs. perpendicular to the pulling direction, only changes in tensile strength. If the polymer chains are anisotropic one would expect to see a considerable difference in the tensile modulus.

The authors are encouraged to provide a rationale for why this is not the case here.

8) Scale bars should be added to images in Figures 7, 9, 11, and 14.

9) The Figure 14c caption says that the 3D printed sphere has a height of 150 um (micrometer), however should this be 150 mm instead?

Reviewer #2 (Remarks to the Author):

The authors already solved all my questions. I would recommend accepting the paper.

Reviewer #3 (Remarks to the Author):

The authors have addressed all my questions successfully.

Responses to Reviewer 1:

Thank you for your valuable suggestions. I have incorporated the required editorial changes in the manuscript and reconsidered the conclusions. Please note that the changes in the revised text are highlighted in red font. All of our responses are given as follows:

1) On lines 46-47 of the revised manuscript, the authors write that “...manufacturing the parts quickly over a period of several to tens of hours.” However, this statement is vague as it does not provide part dimensions associated with the printing time. More effective would be providing a printing rate in terms of curing volume/time.

Response: We agree with your comment and have changed the following sentences to the revised manuscript. The liquid cures at the point of luminous contact, manufacturing the parts by deposition speed up to $10^5 \text{ mm}^3/\text{h}$ ¹⁷.

2) The text in the image of Figure 3B is difficult to read, in part due to the poor color contrast and in-part due to the low resolution and small font size (e.g., scale bars).

Response: We agree with the reviewer’s comment. Fig. 3(b) has been edited to clarify this issue in the revised manuscript.

3) Lines 212-213 the authors state “However, the surface XRD peaks of PR-5CB-0// and PR-5CB-3 \perp ...” – I think the authors meant for this to be PR-5CB-3 for both parallel and perpendicular orientations.

Response: Thank you for identifying that error. This has been corrected in the revised manuscript.

4) Figure 4 key appears to have a similar issue. The final sample in the list should be PR-5CB-3 not PR-5CB-0 as it is currently written.

Response: Thank you for identifying that error. This has been corrected in the revised Fig.4.

5) Line 262: “...5CB, tough nesting and...” should I think be “...5CB, toughness and...”

Response: Thank you for identifying that error. This has been corrected in the revised manuscript.

6) The authors are encouraged to specifically note that the increased crosslink density is due to an increase in physical crosslinks. As the text stands it was not immediately obvious why crosslink density would increase by adding 3% 5CB to the resin as typically in vat

photopolymerization crosslinks refer to covalent bonds as opposed to intermolecular interactions.

Response: We agree with your comment and have changed the following sentences to the revised manuscript.

7) The authors have not sufficiently addressed my concerns with the stress-strain curves in Figure 5a, which show an initially low modulus, followed by a sharp increase. The pre-load force mentioned by the authors in the rebuttal should not account for such a large percentage as that observed for the soft regions in these measurements. Also, preloading is not typically included in the stress-strain plots. The other concern here is that there appears to be no significant difference in tensile modulus for samples oriented parallel vs. perpendicular to the pulling direction, only changes in tensile strength. If the polymer chains are anisotropic one would expect to see a considerable difference in the tensile modulus. The authors are encouraged to provide a rationale for why this is not the case here.

Response: Thank you very much for your query, we are fully aware of the problem. After some discussion and examination, we found that the slack in the previous paragraph was due to the sample not being taut during the tensile test. After improvement we tensed the samples during the test so that the value showed between 0 and preload (preload was set between 0 and 5% of maximum force), then cleared it and started the test again. We reprinted and tested PR-5CB-0//, PR-5CB-0 \perp , PR-5CB-3//, PR-5CB-3 \perp , so on. The results are shown in the modified Fig. 5 and Supplementary Fig.1, where the relaxation in the front part of the stress-strain curve is gone, and also the tensile modulus of the printed samples in both directions shows anisotropy. The final values of tensile strength and elongation at break obtained are not much different from the original results, which have been updated and modified in the manuscript.

8) Scale bars should be added to images in Figures 7, 9, 11, and 14.

Response: Thank you for your recommendation. We have made the necessary changes to Figs. 7, 9, 11 and 14 in the revised manuscript.

9) The Figure 14c caption says that the 3D printed sphere has a height of 150 μm (micrometer), however should this be 150 mm instead?

Response: Thank you for identifying that error. This has been corrected in the revised manuscript.

REVIEWERS' COMMENTS

Reviewer #1 (Remarks to the Author):

The authors have addressed my remaining questions/concerns.